# TaskLoom: Weaving Knowledge Across Tasks in World Models

**Qingzhang Zeng** [1]  **Peixi Peng** [1 2]  **Hang Li** [1]  **Luntong Li** [2]  **Yonghong Tian** [1 2 3]

## Abstract

World models have significantly improved the sample efficiency of model-based reinforcement learning (MBRL) by enabling policy learning in imagination, thereby reducing the need for direct interaction with the real environment. However, most existing world model methods are trained independently for each task or perform multi-task learning using offline datasets, failing to fully exploit the latent relationships among tasks in online interactive scenarios. To address this limitation, we propose TaskLoom, a knowledge-sharing world model architecture for online RL. TaskLoom adopts a grouped two-stage training paradigm: first, the tasks are divided into several groups based on the similarity of world model gradients, and fine-grained knowledge is shared among tasks within each group; second, coarse-grained knowledge is exchanged across groups, enabling hierarchical knowledge transfer and reuse. Experimental results show that TaskLoom outperforms baseline methods on widely used benchmarks such as Proprio Control, Visual Control and Meta-World, validating the effectiveness of the proposed knowledge-sharing mechanism for both low-dimensional state and high-dimensional visual inputs.

## 1. Introduction

Reinforcement learning (RL) has achieved remarkable success in sequential decision-making (Sutton et al., 1998; Silver et al., 2016), but often suffers from poor sample efficiency due to its reliance on extensive environment interaction (Yampolskiy, 2018; Chen et al., 2022; Micheli et al., 2023; Petrenko et al., 2023). Model-based reinforcement

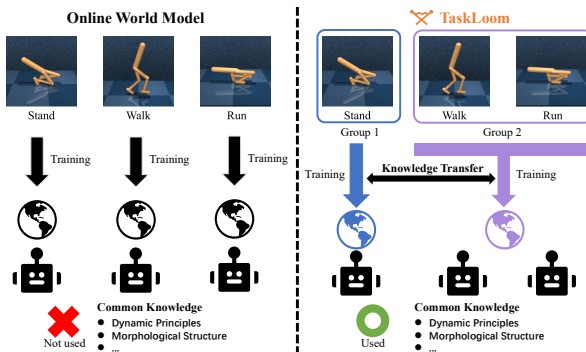

*Figure 1.* Comparison between conventional online world models and TaskLoom. Conventional online world models usually train separate models for each task, neglecting the common knowledge that could be shared across tasks. TaskLoom introduces a hierarchical, group-based knowledge-sharing framework that enables fine-grained sharing within groups and coarse-grained sharing across groups, leveraging inter-task common knowledge to strengthen modeling for individual tasks.

learning (MBRL) mitigates this limitation by learning environment dynamics and leveraging model-based imagination, substantially improving data efficiency (Sutton, 1991).

Among various MBRL approaches, world models have emerged as a particularly effective paradigm for learning environment dynamics and enabling imagination-based policy optimization (Ha & Schmidhuber, 2018; Kaiser et al., 2019; Hafner et al., 2022). World model methods, such as the Dreamer family (Hafner et al., 2019a; 2020; 2025), demonstrate that learning latent dynamics models enables effective policy optimization through imagined rollouts. However, most world model approaches are predominantly designed for single-task learning, where each task is trained with an independent world model. When extended to multi-task settings, this paradigm becomes highly inefficient, as it repeatedly learns similar or identical environment dynamics across tasks. For example, as shown in Figure 1, tasks such as *Walker Stand*, *Walker Walk*, and *Walker Run* in DMControl share identical physical dynamics. Following this line, TD-MPC2 (Hansen et al., 2024) shows that a single world model can scale across large task collections, emphasizing robustness and architectural scalability. However, the

[1]School of Electronic and Computer Engineering, Shenzhen Graduate School, Peking University, China [2]Peng Cheng Laboratory, China [3]National Engineering Research Center of Visual Technology, School of Computer Science, Peking University, China. Correspondence to: Peixi Peng <pxpeng@pku.edu.cn>.

*Proceedings of the $43^{rd}$ International Conference on Machine Learning*, Seoul, South Korea. PMLR 306, 2026. Copyright 2026 by the author(s).

multi-task version of TD-MPC2 is trained on offline multi-task data and does not explicitly address how cross-task knowledge can be exploited in online settings.

Learning a shared world model in online multi-task settings poses several fundamental challenges. In particular, online world model training is inherently unstable and typically requires a large number of interactions to converge, rendering naive multi-task extensions prohibitively expensive. Moreover, unlike offline settings where diverse multi-task data can be aggregated beforehand, online learning must adapt the world model incrementally from non-stationary, task-dependent data streams, increasing the risk of negative transfer and catastrophic interference. Crucially, the model must simultaneously balance task-specific adaptation with cross-task generalization.

To address these limitations, we propose TaskLoom, a world model architecture for online multi-task knowledge sharing. TaskLoom treats the world model as a medium for cross-task knowledge transfer and introduces a structured, two-stage training mechanism to enable hierarchical reuse of environment dynamics. In the first stage, tasks are grouped according to the similarity of their world model gradients, and fine-grained knowledge sharing is performed within each group through joint world model training. In the second stage, TaskLoom explicitly incorporates a coarse-grained knowledge sharing mechanism across groups, facilitating higher-level knowledge transfer beyond group boundaries. Through this staged design, TaskLoom adaptively improves modeling accuracy and sample efficiency for individual tasks in an online learning setting.

Notably, while most online multi-task reinforcement learning (MTRL) methods aim to exploit shared structure across tasks, existing approaches predominantly focus on sharing policies, value functions, or representations, rather than modeling and sharing environment dynamics (Teh et al., 2017; Vithayathil Varghese & Mahmoud, 2020; Reed et al., 2022). In contrast, we provide a framework that enables hierarchical sharing of environment dynamics across tasks, offering a complementary and orthogonal direction to existing MTRL approaches.

Specifically, our main contributions are summarized as follows:

- We introduce a new perspective on leveraging multi-task experience by treating the world model as a form of cross-task knowledge, enabling explicit reuse of environment dynamics to improve individual task learning under an online setting.

- We propose a hierarchical, group-based knowledge sharing mechanism that facilitates both fine-grained collaboration among related tasks and coarse-grained

transfer across task groups through a shared knowledge base.

- We demonstrate through extensive experiments on multiple benchmarks that TaskLoom consistently outperforms baselines under both low-dimensional state inputs and high-dimensional visual observations, and additionally provide auxiliary results to contextualize its performance in online MTRL settings.

## 2. Related Work

### 2.1. Model-based Reinforcement Learning

MBRL (Sutton, 1991) improves sample efficiency by learning a model of environment dynamics and rewards, enabling policy optimization in an imagined space with fewer real interactions. Its performance critically depends on the accuracy of the learned transition and reward models. Early MBRL approaches such as PlaNet (Hafner et al., 2019b) introduced the recurrent state space model (RSSM) to learn compact latent dynamics, significantly reducing data requirements compared to model-free methods. Building on this foundation, the Dreamer series (Hafner et al., 2019a; 2020; 2025) further refined RSSM by incorporating discrete stochastic latent variables and hybrid objective functions, achieving strong performance on both continuous control and visual domains. These advances inspired numerous subsequent studies based on RSSM-style architectures (Lin et al., 2024; Huang et al., 2024; Li et al., 2025; Zhao et al., 2025; Lai et al., 2025). In addition, other MBRL works have explored alternative model architectures and training strategies to enhance robustness and generalization (Micheli et al., 2023; Deng et al., 2023; Zhang et al., 2023; Mattes et al., 2024; Wang et al., 2024; Alonso et al., 2024; He et al., 2025). However, most existing MBRL approaches focus on single-task scenarios and overlook the potential structural similarities and shared dynamics across different tasks. Although some researchers have attempted to explore MBRL in multi-task scenarios (Hansen et al., 2024), they still rely largely on static datasets, and lack mechanisms for continually sharing and adapting world knowledge across multiple tasks in online scenarios.

### 2.2. Multi-task Reinforcement Learning

MTRL aims to jointly train agents on multiple related tasks so that they can acquire both shared knowledge and task-specific strategies, thus improving overall generalization and data efficiency (Teh et al., 2017; Vithayathil Varghese & Mahmoud, 2020). Some works formulate multiple tasks as a conditional sequence generation problem, enabling unified policy learning across tasks (Lee et al., 2022; Reed et al., 2022; Xu et al., 2022; He et al., 2023; Hu et al., 2024). In addition, several studies focus on mitigating gradient

conflicts among tasks to improve the stability of multi-task optimization (Chen et al., 2020; Yu et al., 2020a; Liu et al., 2021). Other approaches seek to capture shared knowledge across tasks through architectural designs (Yang et al., 2020; Sodhani et al., 2021; Sun et al., 2022; D'Eramo et al., 2024; Kong et al., 2025). Building on these ideas, some recent works further introduce task-specific corrections at the policy representation or action level to alleviate negative transfer induced by fully shared policies, while still maintaining parameter sharing (Roberts & Di, 2024; Feng et al., 2025). Nevertheless, most existing methods focus on policy learning or offline settings, while systematic modeling of shared environment dynamics and online knowledge transfer remains underexplored.

### 2.3. Transfer Learning

Transfer learning (TL) in RL leverages knowledge from prior tasks to improve learning efficiency and generalization (Zhu et al., 2023). Many existing approaches focus on reusing learned representations or network components to enable knowledge transfer across tasks, including modular architectures and disentangled representations (Devin et al., 2017; Andreas et al., 2017; Zhang et al., 2018; Rusu et al., 2016; Fernando et al., 2017; Barreto et al., 2017; 2018; Borsa et al., 2019). However, these methods typically assume a sequential source–target setting and do not explicitly model shared environment dynamics or support concurrent multi-task learning. In contrast, our method performs simultaneous multi-task learning via a shared world model that captures and transfers environment structure across tasks.

## 3. Preliminary

### 3.1. Reinforcement Learning

Reinforcement learning (RL) is commonly formalized as a Markov Decision Process (MDP) (Bellman, 1957). A standard MDP can be represented as a five-tuple $(\mathcal{S}, \mathcal{A}, \mathcal{T}, \mathcal{R}, \gamma)$, where $\mathcal{S}$ denotes the state space and $\mathcal{A}$ denotes the action space. The transition probability function $\mathcal{T} : \mathcal{S} \times \mathcal{A} \times \mathcal{S} \rightarrow [0, 1]$ describes the environment dynamics through the probability distribution $p(s_{t+1} \mid s_t, a_t)$. The reward function $\mathcal{R} : \mathcal{S} \times \mathcal{A} \rightarrow \mathbb{R}$ evaluates the immediate return obtained by the agent when taking an action $a_t$ in a given state $s_t$. The discount factor $\gamma \in [0, 1]$ balances short-term and long-term rewards.

The objective of RL is to learn an optimal policy $\pi^*$ that maximizes the expected cumulative discounted reward:

$$\pi^* = \arg\max_{\pi} \mathbb{E}_{\pi} \left[ \sum_{t=0}^{\infty} \gamma^t \mathcal{R}(s_t, a_t) \right]. \tag{1}$$

In partially observable environments, the agent receives partial observations instead of the true state. This is formalized as a Partially Observable Markov Decision Process (POMDP) (Sutton et al., 1998), defined as a tuple $(\mathcal{S}, \mathcal{A}, \mathcal{O}, \mathcal{T}, \mathcal{R}, O, \gamma)$, where $\mathcal{O}$ denotes the observation space and $O : \mathcal{S} \times \mathcal{O} \rightarrow [0, 1]$ specifies the observation probability distribution $p(o_t \mid s_t)$. In this setting, the agent aims to learn a policy $\pi(a_t \mid o_t)$ that maximizes the long-term return based on limited perceptual information.

### 3.2. Recurrent State Space Model

The recurrent state space model (RSSM) (Hafner et al., 2019b) combines deterministic hidden states with stochastic latent variables, allowing the model to capture environmental uncertainty while maintaining long-term temporal memory. This hybrid representation enables informative latent dynamics that support reasoning and planning in MBRL.

The RSSM-based world model consists of an encoder, a recurrent sequence model, several predictors, and a decoder. At each time step, the deterministic hidden state is updated by a **sequence model** $h_t = f_\phi(h_{t-1}, z_{t-1}, a_{t-1})$. The **encoder** then infers a stochastic latent variable $z_t \sim q_\phi(z_t \mid h_t, o_t)$ from the observation. The hidden state $h_t$ is also used by a **dynamics predictor** to predict a prior latent state without access to the current observation, $\hat{z}_t \sim p(\hat{z}_t \mid Dyn_\phi(h_t))$, where $p$ is a categorical distribution, and $Dyn_\phi(h_t)$ is used to calculate the specific distribution parameters from $h_t$. For clarity, we refer to $Dyn$ as the dynamics predictor in our method. Given $(h_t, z_t)$, the model further employs additional **predictors** to estimate the reward $\hat{r}_t \sim p_\phi(\hat{r}_t \mid h_t, z_t)$, the continuation signal $\hat{c}_t \sim p_\phi(\hat{c}_t \mid h_t, z_t)$, and a **decoder** to reconstruct the observation $\hat{o}_t \sim p_\phi(\hat{o}_t \mid h_t, z_t)$. Note $\phi$ is the network parameter, and is removed for simplicity hereinafter.

The training objective of RSSM comprises three major loss components: the prediction loss $\mathcal{L}_{\text{pred}}$, the dynamics loss $\mathcal{L}_{\text{dyn}}$, and the representation loss $\mathcal{L}_{\text{rep}}$. The model learns accurate and generalizable latent dynamics by minimizing the combined objective with corresponding loss weights $\beta_{\text{pred}} = 1$, $\beta_{\text{dyn}} = 1$, and $\beta_{\text{rep}} = 0.1$:

$$\mathcal{L}(\phi) = \mathbb{E}_{q_\phi} \left[ \sum_{t=1}^{T} \left( \beta_{\text{pred}} \mathcal{L}_{\text{pred}} + \beta_{\text{dyn}} \mathcal{L}_{\text{dyn}} + \beta_{\text{rep}} \mathcal{L}_{\text{rep}} \right) \right]. \tag{2}$$

DreamerV3 (Hafner et al., 2025) further stabilizes training via free bits and KL balancing to mitigate posterior collapse, and employs a symlog transformation for scale-invariant normalization. Through RSSM, the agent can perform imagination rollouts in latent space, enabling efficient model-based policy optimization.

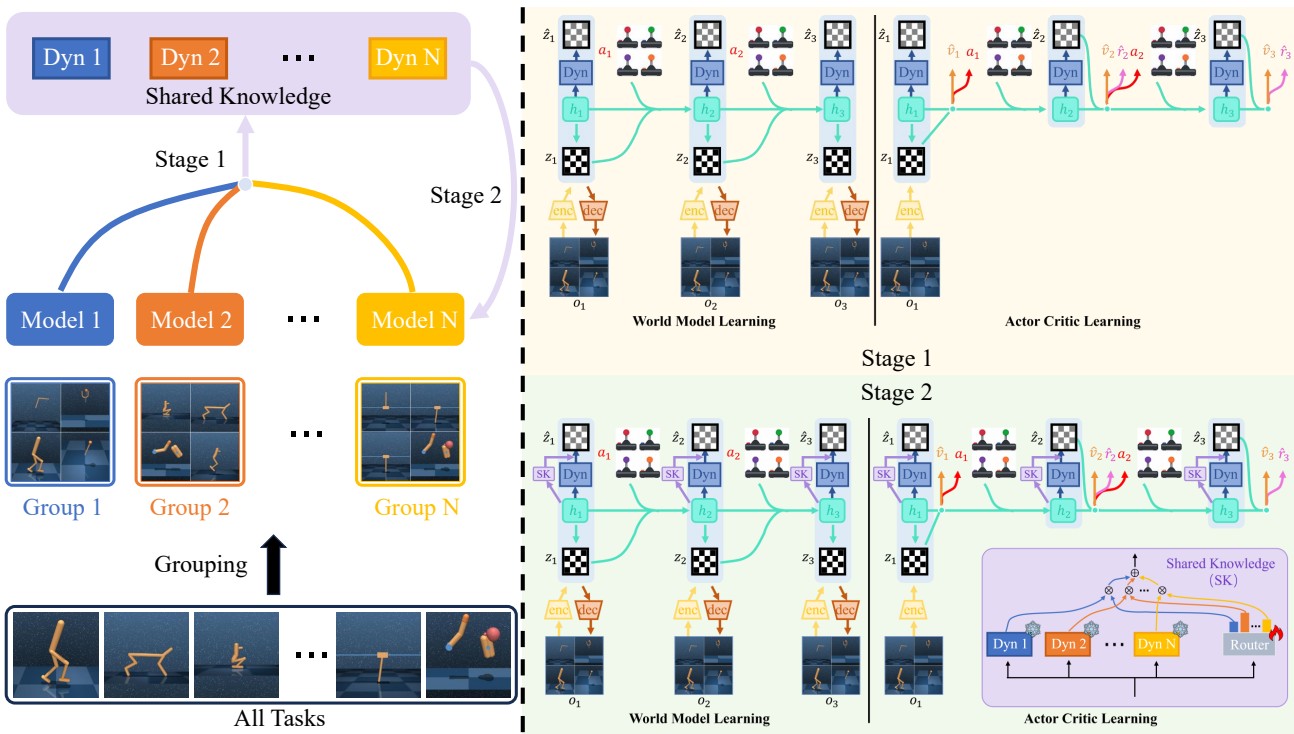

*Figure 2.* Overview of TaskLoom. All tasks are first grouped based on gradient similarity. **Stage 1:** Tasks within each group are trained collaboratively under a shared world model, while each task retains its own actor and critic networks. **Stage 2:** The Shared Knowledge ($SK$) module is introduced on top of the Stage 1 model to facilitate inter-group knowledge transfer. For each group, its corresponding world model and the router network in $SK$ are updated continually.

## 4. Method

In this section, we propose TaskLoom, a multi-task knowledge-sharing world model architecture. A central challenge in multi-task world modeling lies in how to share knowledge across tasks while mitigating negative transfer, as tasks may vary substantially in their dynamics. TaskLoom addresses this challenge by structuring tasks according to their similarity and organizing knowledge sharing at different granularities. Specifically, built upon DreamerV3 (Hafner et al., 2025), we adopt a grouped two-stage training framework that enables effective knowledge reuse among similar tasks (intra-group), while still allowing more abstract world knowledge to be transferred across dissimilar ones (inter-group), as illustrated in Figure 2.

### 4.1. Gradient-Based Grouping and Agglomerative Clustering

Our task grouping strategy adopts a multi-step gradient-based grouping followed by agglomerative clustering. In each gradient-based grouping step, we first compute agreement vectors (Hu et al., 2024) for every task based on the current model to quantify task similarity, and then apply

K-means clustering to partition tasks into groups (Kong et al., 2025). However, since the grouping results obtained through agent–environment interactions and model gradient updates are often unstable—and given that gradients in non-linear networks depend not only on the input but also on the model state—we further employ agglomerative clustering to consolidate the grouping results obtained over training.

Concretely, for a set of tasks that have not yet been grouped, we first perform a dedicated grouping run. Using the task-shared RSSM, all $K$ tasks are jointly trained. During this run, we perform gradient-based grouping $M$ times at fixed training intervals, yielding a sequence of clustering snapshots $\{C^{(m)}\}_{m=1}^{M}$. Here, $C^{(m)}(i)$ denotes the group assignment of task $i \in \{1, \ldots, K\}$ at the $m$-th grouping step. To aggregate these potentially noisy grouping results, we construct a weighted co-association matrix:

$$A_{ij} = \frac{\sum_{m=1}^{M} w_m \cdot \mathbf{1}\big(C^{(m)}(i) = C^{(m)}(j)\big)}{\sum_{m=1}^{M} w_m}. \quad (3)$$

Since grouping results are expected to become more reliable as training progresses, we assign linearly increasing weights

$w_m$ to clustering snapshots, ranging from 1 to 2. We found that the final grouping result is insensitive to this weight range, and this simple scheme is adopted for clarity and interpretability.

We then perform agglomerative clustering (Müllner, 2011) on the co-association matrix to obtain the final task grouping. Specifically, $A$ is treated as a similarity matrix and converted to a distance matrix $D = 1 - A$. Agglomerative clustering with average linkage is applied to $D$, and the hierarchy is cut to produce $N$ clusters, corresponding to $N$ task groups.

This grouping approach inherently mitigates instability; thus, we only need to perform grouping once for ungrouped tasks. The resulting task partitions can be reused in all subsequent experiments without rerunning the grouping process. By dividing $K$ tasks into $N$ groups ($N < K$), the number of required world models is effectively reduced from $K$ (per-task training) to $N$ (per-group training).

### 4.2. Intra-Group Collaborative Training

Unlike existing world model methods, which maintain a separate world model for each task, our approach requires only one shared world model per group. This work focuses on the shared knowledge mechanism of world models, hence actor–critic networks remain task-specific. We will explore the multi-task version actor–critic in the future.

In our framework, each training batch consists of data sampled from all tasks within a group, rather than from a single task. Specifically, suppose a group contains $K_{\text{group}}$ tasks. At step $t$, the training data from task $k \in \{1, 2, \ldots, K_{\text{group}}\}$ is denoted as $\mathbf{X}_k^{(t)}$. We treat the union of samples from all tasks in the group as a single training batch:

$$\mathbf{X}^{(t)} = \bigcup_{k=1}^{K_{\text{group}}} \mathbf{X}_k^{(t)}. \tag{4}$$

Since intra-group tasks are highly similar—and the grouping itself is based on gradient similarity—gradient conflicts are minimized. Consequently, joint training within the group encourages the model gradients to evolve in a more consistent and coherent direction.

Analogous to human cognitive transfer (e.g., learning by analogy), we posit that when the model learns one task, it simultaneously captures fine-grained patterns or rules that generalize across similar tasks. These shared micro-knowledge components enable the model to learn related tasks more efficiently. Our proposed architecture achieves this fine-grained knowledge sharing through the shared world model. Although the fine-grained knowledge learned from one task may not perfectly fit other tasks within the same group, the model's shared structure allows such knowledge to be refined through mutual influence, reducing overfitting and

enhancing generalization across related tasks.

### 4.3. Inter-Group Collaborative Training

While inter-group tasks may differ significantly in gradient behavior, many of them share latent common rules or dynamics—for example, "obstacles should be avoided," or "walkers share the same physical structure regardless of whether the task is stand or walk." Furthermore, observations from RSSM indicate that combining deterministic states with sampled stochastic states improves performance. We interpret the stochastic state as imagination or hypothesis, analogous to how humans infer possible outcomes beyond direct perception. For instance, when we see a stationary ball on a table, we can imagine it rolling if perturbed—but not transforming into a bird and flying away. Such imagination is guided by common-sense world knowledge.

Hence, we view predicting stochastic states as implicitly capturing coarse-grained, abstract world knowledge. Consequently, we introduce a shared knowledge ($SK$) base built upon the group-level dynamics predictors learned in Stage 1. We instantiate $SK$ as a mixture-of-experts (MoE) model. For group $g$, each expert corresponds to the dynamics predictor $Dyn_i$ of a different task group $i \neq g$, and a lightweight MLP router dynamically aggregates their outputs. Importantly, the dynamics predictor $Dyn_g$ of the current group is explicitly excluded from $SK$, ensuring that $SK$ provides only cross-group, coarse-grained information and does not interfere with the fine-grained knowledge already captured within the group. Formally, $SK$ is defined as:

$$SK(h_t) = \sum_{i \neq g} \alpha_i(h_t) \, Dyn_i(h_t), \tag{5}$$

where the routing weights are computed as:

$$\alpha_i(h_t) = \text{Softmax}(Router(h_t))_i. \tag{6}$$

During this stage, the expert dynamics predictors $\{Dyn_i\}_{i \neq g}$ are fixed in the shared knowledge branch, and only the router network $Router$ is trained; all other components follow the same training scheme as in Stage 1.

In our method, we extend dynamics predictor of RSSM. Specifically, for each group $g$, the group-specific dynamics predictor $Dyn_g(h_t)$ provides the primary, fine-grained prediction, while the shared knowledge base $SK(h_t)$ contributes a residual correction encoding coarse-grained, cross-group world knowledge. Then, the stochastic latent state $\hat{z}_t$ is sampled as:

$$\hat{z}_t \sim p(\hat{z}_t | Dyn_g(h_t) + SK(h_t)). \tag{7}$$

where $\sim p$ keep consistent with RSSM dynamics predictor.

This residual form ensures that group-specific dynamics remain the dominant source of information, while shared knowledge only biases the latent distribution toward abstract yet plausible outcomes shared across task groups. Through this design, the model incorporates abstract world knowledge shared across groups while preserving the specialization of each group.

Due to page limitations, the pseudocode for the training procedure is detailed in Appendix A.

## 5. Experiment

We evaluate our method on two mainstream benchmarks to comprehensively assess its effectiveness in both single-task and multi-task RL settings.

**DMControl Benchmark.** We first evaluate our method on the widely used DMControl benchmark (Tassa et al., 2018) to demonstrate that sharing world model knowledge across tasks can consistently improve performance on individual tasks. To assess robustness under different observation modalities, we consider two experimental settings: Proprio Control, where agents receive low-dimensional proprioceptive state vectors, and Visual Control, where agents learn directly from high-dimensional image observations. Following the official code of DreamerV3 (Hafner et al., 2025), we conduct a systematic evaluation on 18 Proprio Control tasks and 20 Visual Control tasks respectively. These tasks cover a diverse range of domains including classical locomotion and robotic manipulation. They span both dense- and sparse-reward settings, providing a comprehensive testbed for evaluating generalization and learning efficiency. To highlight training efficiency, all methods are evaluated with only 50K environment interaction steps per task. Unless otherwise specified, results are averaged over 5 seeds.

We compare our approach against a set of widely used and competitive baseline methods, including both model-free and model-based algorithms: PPO (Schulman et al., 2017), DMPO (Abdolmaleki et al., 2018), CURL (Laskin et al., 2020), DrQ-V2 (Yarats et al., 2022), TD-MPC2 (Hansen et al., 2024), and DreamerV3 (Hafner et al., 2025). The results are from (Hafner et al., 2025), except TD-MPC2 is re-implemented by the official code. In addition, we include a comparison with an offline multi-task variant of TD-MPC2, denoted as TD-MPC2*. TD-MPC2* is trained on a dataset containing 345M transitions (i.e., environment steps), collected from 30 DMControl tasks. We select the version of TD-MPC2* which contains similar network parameters with us (19M vs. 21M). Due to the substantially different training setups, results from TD-MPC2* are provided for reference only. Owing to data availability constraints, we do not include TD-MPC2* in the Visual Control setting.

**Meta-World Multi-Task Benchmark.** We further evalu-

*Table 1.* Performance on the Proprio Control benchmark demonstrates that TaskLoom can effectively learn from proprioceptive inputs. For each task, the best-performing online method is highlighted in **bold**, and the second-best is underlined. Full results on the Proprio Control benchmark are provided in Appendix C.

| Algorithm | Mean | Median |
|---|---|---|
| Offline (345M Transitions) | | |
| TD-MPC2* | 650.7 | 800.2 |
| Online (50k Interaction Steps) | | |
| PPO | 50.0 | 17.6 |
| DMPO | 331.7 | 224.4 |
| TD-MPC2 | 619.1 | 750.6 |
| DreamerV3 (18×0.25 GPU Days) | 456.1 | 501.5 |
| **TaskLoom (Ours)** (1.3 GPU Days) | **647.4** | **775.7** |

ate our method on Meta-World (Yu et al., 2020b), a widely adopted benchmark for MTRL. While our method is primarily designed for comparison against single-task baselines, we include results on Meta-World as a reference point for online MTRL. Following (Roberts & Di, 2024), we consider the MT10 and MT50 benchmarks, which comprise 10 and 50 manipulation tasks executed by a simulated Sawyer robot arm. Each task is defined over a 12-dimensional state space, including the 3D Cartesian position of the end effector, the 3D positions of one or two objects, and a target position vector. To show the efficiency of our method, we evaluate all algorithms within 150k environment steps (Feng et al., 2025) and report results over 5 seeds.

We compare our method with representative online MTRL baselines, including MT-SAC (Yu et al., 2020b), Soft Modularization (Yang et al., 2020), CARE (Sodhani et al., 2021), CARE+PTSL (Roberts & Di, 2024), and TSAC (Feng et al., 2025). We note that most of these baselines are model-free approaches, whereas our method represents a model-based approach to MTRL. Therefore, this comparison is intended to provide a reference for understanding the relative performance of world-model-based multi-task learning, rather than a direct contradiction to existing model-free methods. All selected baselines either report results on chosen benchmarks or provide publicly available implementations, ensuring a fair and reproducible comparison.

More experiment details are shown in Appendix B.

### 5.1. Comparison on DMControl

**Proprio Control.** The results for Proprio Control are shown in Table 1, with training curves provided in Appendix C. TaskLoom achieves performance that is comparable to or better than existing single-task methods on most tasks, with a mean and median return of 647.4 and 775.7, respectively. This substantially outperforms DreamerV3 (456.1 / 501.5) and TD-MPC2 (619.1 / 750.6). These results indicate that

*Table 2.* Performance on the Visual Control benchmark demonstrates that TaskLoom can effectively learn from visual inputs. The best result in each column is highlighted in **bold**, while the second-best result is underlined. Full results on the Visual Control benchmark are provided in Appendix C.

| Algorithm | Mean | Median |
|---|---|---|
| PPO | 55.2 | 16.0 |
| CURL | 349.1 | 228.7 |
| DrQ-V2 | 267.7 | 165.4 |
| TD-MPC2 | 396.7 | 314.2 |
| DreamerV3 | 382.0 | 297.8 |
| TaskLoom (Ours) | **455.3** | **464.3** |

introducing cross-task knowledge sharing during training enables the learned world model to more effectively support control on individual tasks.

Consistent performance gains are observed across a diverse set of tasks (see Appendix C, Table 10), including sparse-reward settings (e.g., *Cartpole Swingup Sparse*) and dynamically challenging control tasks (e.g., *Reacher Hard*). This suggests that, despite differences in task objectives and reward structures, these tasks share reusable knowledge at the level of dynamics modeling and representation learning. Such shared structure is not explicitly exploited by standard single-task world model training paradigms such as DreamerV3. Importantly, TaskLoom does not simply jointly train a single model across all tasks. Instead, it adopts a grouped two-stage training scheme that enables effective cross-task knowledge sharing while preserving task-specific characteristics, thereby mitigating the negative transfer that may arise from naive parameter sharing.

**Visual Control.** The results for Visual Control are shown in Table 2, with training curves provided in Appendix C. Compared to Proprio Control, Visual Control tasks rely on high-dimensional image observations, posing greater challenges for representation learning and world model construction. Under this more demanding setting, TaskLoom still achieves the best overall performance, with a mean and median return of 455.3 and 464.3, respectively. This clearly outperforms DreamerV3 (382.0 / 297.8) and the strong baseline TD-MPC2 (396.7 / 314.2). These results demonstrate that even under significantly increased perceptual complexity, world models learned via cross-task knowledge sharing can effectively enhance control performance.

Across multiple visual control tasks, TaskLoom exhibits improvements over single-task methods (see Appendix C, Table 11), suggesting that cross-task knowledge sharing helps alleviate the learning instability caused by the tight coupling between high-dimensional visual representations and dynamics modeling. Meanwhile, on relatively simple tasks or tasks with well-shaped reward structures, TaskLoom performs on par with the best single-task methods, indicat-

*Table 3.* Success rate of baselines and our method on Meta-World.

| Algorithm | MT10 | MT50 |
|---|---|---|
| MT-SAC | 0.294 ± 0.092 | 0.179 ± 0.043 |
| SoftMod | 0.243 ± 0.092 | 0.122 ± 0.058 |
| CARE | 0.301 ± 0.175 | 0.291 ± 0.050 |
| CARE+PTSL | 0.438 ± 0.087 | 0.318 ± 0.037 |
| TSAC | 0.390 ± 0.115 | 0.362 ± 0.051 |
| TaskLoom (Ours) | **0.517 ± 0.075** | **0.470 ± 0.042** |

ing that the benefits of cross-task sharing are naturally limited when a single task alone already supports stable world model learning. Notably, on challenging visual control tasks such as *Reacher Hard*, single-task methods often fail to learn effective policies reliably. In contrast, by sharing dynamics and visual representation knowledge across tasks, TaskLoom substantially outperforms DreamerV3 (466.0 vs. 53.8), demonstrating its effectiveness in challenging visual and dynamical settings.

**Summary.** Overall, the results on both Proprio Control and Visual Control validate the effectiveness of TaskLoom's grouped two-stage training framework. Across low-dimensional state inputs and high-dimensional visual observations, as well as sparse and dense reward settings, TaskLoom consistently demonstrates strong performance. By learning more robust world models through intra-group collaborative training and enabling inter-group knowledge transfer via shared knowledge, TaskLoom effectively exploits common structure across tasks while mitigating negative transfer, resulting in superior performance compared to single-task world model approaches.

### 5.2. Comparison on Meta-World

As shown in Table 3, our method achieves the highest success rates on both MT10 and MT50 benchmarks, with $0.517 \pm 0.075$ and $0.470 \pm 0.042$, respectively, consistently outperforming representative online MTRL baselines such as MT-SAC and CARE+PTSL. These results provide a reference indicating that our framework maintains competitive performance when applied to more challenging multi-task settings. Training curves are provided in Appendix C.

### 5.3. Ablation Study

We conduct a comprehensive ablation study on Proprio Control to validate the effectiveness of the key design components of TaskLoom. For clarity, we define the following model variants evaluated in our ablation experiments: (1) **Shared-Dreamer (Baseline)**, where no task grouping is applied and all tasks are jointly trained with a single shared world model; (2) **G-Baseline**, which introduces task grouping and trains different groups separately in Stage 1 while disabling Stage 2; (3) **G-Baseline+SK (Ours)**, which

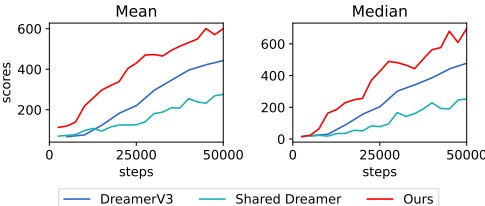

*Figure 3.* Performance comparison among DreamerV3, Shared-Dreamer, and our method. For each method, results are first aggregated across all seeds for each task, and the mean and median are then computed across all tasks. Direct multi-task joint training based on DreamerV3 (Shared-Dreamer) results in performance degradation, whereas our method achieves stronger performance.

*Table 4.* Ablation study results on the Proprio Control benchmark. Full ablation results on the Proprio Control benchmark are provided in Appendix D.

| Algorithm | Mean | Median |
|---|---|---|
| DreamerV3 | 456.1 | 501.5 |
| Shared-Dreamer | 280.3 | 257.0 |
| G-Baseline | 556.8 | 533.4 |
| G-Baseline+SK (Ours) | **647.4** | **775.7** |
| G-Baseline+SK-Dyn (Replacement) | 604.5 | 704.7 |

further introduces a shared knowledge ($SK$) base; and (4) **G-Baseline+SK-Dyn (Replacement)**, which incorporates the dynamics predictor of the current group into the shared knowledge base, thereby replacing the original group-specific dynamics predictor for the current group.

**Limitations of Direct Multi-Task Joint Training.** As shown in Figure 3, a comparison between DreamerV3 and Shared-Dreamer reveals that naively training all tasks jointly within a single world model leads to a substantial degradation in overall performance. This result indicates that, without an appropriate structural design, sharing a world model across tasks not only fails to effectively exploit cross-task knowledge but instead introduces negative transfer that harms performance. The underlying reason is that employing a fully shared world model across heterogeneous tasks implicitly assumes a high degree of consistency across tasks in terms of dynamics and reward distributions. In practice, however, different tasks often exhibit significant discrepancies in dynamics scales, control frequencies, and reward sparsity. Such heterogeneity makes simple joint training prone to negative transfer, causing the learned world model to deteriorate on certain tasks. This observation highlights that effectively extracting and leveraging shared knowledge across diverse tasks is itself a non-trivial challenge, which constitutes the core motivation behind TaskLoom.

**Effectiveness of Task Grouping in Stage 1.** As reported in Table 4, comparing Shared-Dreamer with G-Baseline

demonstrates the effect of introducing task grouping within the Stage 1 training framework. Specifically, G-Baseline achieves an average return of 556.8, representing an increase of 276.5 points over Shared-Dreamer. This improvement not only compensates for the performance loss caused by naive multi-task training but also surpasses DreamerV3 (456.1) by 100.7 points on average. By enabling fine-grained knowledge sharing within each group, the model can better capture task correlations and significantly improve overall performance, effectively mitigating the negative transfer introduced by indiscriminate parameter sharing.

Nevertheless, we observe that relying solely on intra-group knowledge sharing still has limitations. For instance, on the *Hopper Hop* task, G-Baseline performs worse (0.6) than Shared-Dreamer (3.9) (see Appendix D, Table 13). This phenomenon suggests that completely isolating knowledge flow across task groups prevents certain tasks from benefiting from general dynamics or control knowledge present in other groups, thereby limiting further performance gains.

**Overall Benefits of Multi-Level Knowledge Sharing.** As shown in Table 4, the full model (Ours), which incorporates inter-group shared knowledge, achieves the best performance among all methods. It attains an average return of 647.4, improving upon G-Baseline by 90.6 points and outperforming DreamerV3 by 191.3 points. Moreover, the median return is significantly increased to 775.7, ranking highest across all compared approaches.

**Necessity of Decoupled Group-Specific Dynamics.** We compare the full model with G-Baseline+SK-Dyn (Replacement), where group-specific dynamics are absorbed into the shared knowledge base. As shown in Table 4, this replacement leads to a clear performance degradation, with the mean return dropping from 647.4 to 604.5. This suggests that overloading the shared knowledge base with group-specific dynamics weakens its ability to capture transferable structure. In contrast, explicitly decoupling group-specific dynamics, as in TaskLoom, better balances generalization and specialization.

## 6. Conclusion

We propose TaskLoom, a grouped two-stage world model framework for structured and efficient multi-task knowledge sharing. By combining fine-grained intra-group sharing with inter-group knowledge transfer, TaskLoom improves learning efficiency while retaining task-specific policies. TaskLoom consistently outperforms strong single-task baselines across proprioceptive and visual control settings, showing robustness to input modalities and scalability to high-dimensional observations. Auxiliary multi-task experiments suggest that TaskLoom maintains competitive performance when applied to online MTRL settings. Overall, TaskLoom

highlights the benefits of structured multi-stage knowledge sharing in world model learning and provides a practical direction toward more generalizable MBRL.

## Acknowledgements

The study was funded by the Shenzhen Science and Technology Program (KQTD20240729102051063), the National Natural Science Foundation of China under contracts No. 62422602, No. 62372010, No. 62425101, No. 62332002, No. 62372010, No. 62206281, Key Laboratory Grants 241-HF-D05-01, and the major key project of the Peng Cheng Laboratory (PCL2021A13 and PCL2025A02). Computing support was provided by Pengcheng Cloudbrain.

## Impact Statement

This paper presents work whose goal is to advance the field of Machine Learning. There are many potential societal consequences of our work, none which we feel must be specifically highlighted here.

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

## A. Pseudocode

The training pseudocode is summarized in Algorithm 1.

## B. Experiment Details

This section contains experiment details.

### B.1. Training Details

Using the 18 tasks from Proprio Control as an example, under the setting where all tasks are trained in parallel, our method requires approximately one day of training per seed in total for all tasks, after an additional 0.3 days for task grouping, on NVIDIA GeForce RTX 4090 GPUs. In terms of memory consumption, each group occupies about 2000 MB of GPU memory in Stage 1, and approximately 3000 MB per group in Stage 2.

### B.2. Task grouping

Clustering snapshots used for agglomerative clustering are obtained via gradient-based grouping over a total of 12500 interaction steps, at which point the final grouping results are observed to stabilize. For simplicity, we set the number of groups to half of the total number of tasks by default and generate the grouping accordingly.

The detailed grouping of tasks used in our experiments are shown in Table 5, Table 6, Table 7 and Table 8.

*Table 5.* Task grouping for the Proprio Control benchmark.

| Group | Tasks |
|---|---|
| Group 0 | Reacher Easy, Reacher Hard |
| Group 1 | Acrobot Swingup, Cup Catch, Pendulum Swingup |
| Group 2 | Finger Spin, Finger Turn Easy, Finger Turn Hard |
| Group 3 | Cartpole Balance Sparse, Cartpole Swingup Sparse |
| Group 4 | Walker Run, Walker Walk |
| Group 5 | Cartpole Balance, Cartpole Swingup |
| Group 6 | Hopper Hop, Hopper Stand |
| Group 7 | Cheetah Run |
| Group 8 | Walker Stand |

*Table 6.* Task grouping for the Visual Control benchmark.

| Group | Tasks |
|---|---|
| Group 0 | Finger Spin, Finger Turn Easy, Finger Turn Hard |
| Group 1 | Walker Run, Walker Stand, Walker Walk |
| Group 2 | Reacher Easy, Reacher Hard |
| Group 3 | Cartpole Balance, Cartpole Balance Sparse, Cartpole Swingup, Cartpole Swingup Sparse |
| Group 4 | Hopper Hop, Hopper Stand |
| Group 5 | Pendulum Swingup |
| Group 6 | Acrobot Swingup |
| Group 7 | Cup Catch |
| Group 8 | Cheetah Run |
| Group 9 | Quadruped Run, Quadruped Walk |

### B.3. Hyperparameters

Hyperparameters are shown in Table 9.

*Table 7.* Task grouping for the MT10 benchmark.

| Group | Tasks |
|---|---|
| Group 0 | Button Press Topdown, Drawer Close, Drawer Open, Peg Insert Side, Push |
| Group 1 | Window Open |
| Group 2 | Door Open, Window Close |
| Group 3 | Pick Place |
| Group 4 | Reach |

*Table 8.* Task grouping for the MT50 benchmark.

| Group | Tasks |
|---|---|
| Group 0 | Pick Place Wall, Shelf Place |
| Group 1 | Push, Push Wall |
| Group 2 | Button Press, Button Press Wall |
| Group 3 | Reach, Reach Wall |
| Group 4 | Disassemble, Pick Out Of Hole, Push Back |
| Group 5 | Button Press Topdown, Button Press Topdown Wall, Door Unlock |
| Group 6 | Hammer, Handle Pull, Lever Pull |
| Group 7 | Drawer Close, Handle Press |
| Group 8 | Coffee Pull, Plate Slide Back |
| Group 9 | Basketball, Plate Slide Side, Plate Slide, Soccer |
| Group 10 | Coffee Button, Door Close, Handle Press Side, Handle Pull Side, Plate Slide Back Side |
| Group 11 | Sweep Into, Sweep |
| Group 12 | Peg Insert Side |
| Group 13 | Drawer Open |
| Group 14 | Box Close, Window Close |
| Group 15 | Door Lock, Door Open |
| Group 16 | Hand Insert |
| Group 17 | Peg Unplug Side, Pick Place |
| Group 18 | Coffee Push |
| Group 19 | Dial Turn |
| Group 20 | Bin Picking |
| Group 21 | Assembly |
| Group 22 | Faucet Close, Faucet Open |
| Group 23 | Window Open |
| Group 24 | Stick Pull, Stick Push |

## C. Training Results

Full results on DMControl are shown in Table 10 and Table 11. Training curves are shown in Figure 4, Figure 5, Figure 6 and Figure 7.

### C.1. Evaluation on Robotics Tasks

To further evaluate the effectiveness of our two-stage method in complex robotics domains, we conduct additional experiments on the Meta-World benchmark. As shown in Table 12, our approach significantly outperforms both the standard DreamerV3 and the Shared-Dreamer baselines. These results demonstrate that our two-stage mechanism remains highly effective and yields substantial performance gains even in challenging robotics environments, successfully mitigating negative transfer.

## D. Ablation

Full ablation results discussed in the main body are shown in Table 13.

*Table 9.* TaskLoom hyperparameters.

| Name | Symbol | Value |
|---|---|---|
| **General** | | |
| Stage 2 start step | – | $totalsteps \times 0.8$ |
| Replay capacity | – | $10^6$ |
| Batch size | $B$ | 4 |
| Batch length | $T$ | 64 |
| Activation | – | RMSNorm + SiLU |
| Update-to-data ratio | – | 1.0 |
| **World Model** | | |
| Reconstruction loss scale | $\beta_{\text{pred}}$ | 1 |
| Dynamics loss scale | $\beta_{\text{dyn}}$ | 0.5 |
| Representation loss scale | $\beta_{\text{rep}}$ | 0.1 |
| Latent unmix | – | 1% |
| Free nats | – | 1 |
| Learning rate | – | $1 \times 10^{-4}$ |
| **Actor Critic** | | |
| Imagination horizon | $H$ | 15 |
| Return lambda | $\lambda$ | 0.95 |
| Learning rate | – | $3 \times 10^{-5}$ |

### D.1. Scaling the Total Number of Tasks

To investigate the performance implications of scaling the total number of tasks, we conduct an ablation study within the Proprio Control setting. We train our model using varying total task counts (6, 12, and 18 tasks). Specifically, these configurations correspond to the first 6, the first 12, and all 18 tasks from the following ordered list: Acrobot Swingup, Cup Catch, Pendulum Swingup, Finger Spin, Finger Turn Easy, Finger Turn Hard (which serve as our fixed subset of 6 base tasks), Reacher Easy, Reacher Hard, Cartpole Balance Sparse, Cartpole Swingup Sparse, Cartpole Balance, Cartpole Swingup, Walker Run, Walker Walk, Hopper Hop, Hopper Stand, Cheetah Run, and Walker Stand. We track the performance of the fixed 6 base tasks across all three settings.

As shown in Table 14, the performance on the base tasks consistently improves as the total number of training tasks increases. This evidence clearly indicates that our method effectively scales up and positively leverages a broader diversity of training tasks to enhance overall performance.

### D.2. Performance under Extended Training Horizon

To investigate the performance implications of a longer interaction budget and ensure that our observed gains reflect genuinely improved asymptotic performance rather than merely faster initial convergence, we extend the training horizon on the Proprio Control benchmark to 250K steps (exactly 5× our original 50K budget).

As shown in Table 15, our method not only sustains its clear advantage under this 5× longer horizon but continues to consistently outperform strong baselines, including DreamerV3 and TD-MPC2. This substantial extension provides concrete evidence that our approach successfully mitigates negative transfer and achieves superior long-run performance, effectively addressing potential concerns regarding sub-optimal asymptotic convergence.

### D.3. Automated Gradient-Based Grouping vs. Intuitive Manual Grouping

To demonstrate the necessity and advantage of our automated gradient-based grouping mechanism over simple human intuition, we evaluate an Intuitive Manual Grouping baseline on the Proprio Control benchmark. The core objective of our gradient-based mechanism is to provide an automated, scalable, general, and highly effective pipeline. While it may capture certain intuitive similarities in simpler benchmarks, its true value lies in extracting underlying dynamic synergies directly from data, completely removing the bottleneck of human bias and the need for prior domain knowledge.

In the Intuitive Manual Grouping baseline, tasks are grouped strictly according to their shared embodiment (e.g., placing all Cartpole tasks in one group and all Walker tasks in another, exactly as human intuition dictates). As shown in Table 16, our

*Table 10.* Full performance on the Proprio Control benchmark demonstrates that TaskLoom can effectively learn from proprioceptive inputs. For each task, the best-performing online method is highlighted in **bold**, and the second-best is underlined. While TD-MPC2* is trained offline on a dataset of 345M transitions covering 30 DMControl tasks, our method achieves comparable performance within only 50k online interaction steps.

| Task | PPO | DMPO | TD-MPC2 | DreamerV3 | TaskLoom (Ours) | TD-MPC2* |
|---|---|---|---|---|---|---|
| Acrobot Swingup | 8.5 | 53.6 | **182.7** | 30.5 | 173.0 | 982.0 |
| Cartpole Balance | 335.0 | 991.7 | 964.2 | 839.7 | **992.9** | 151.5 |
| Cartpole Balance Sparse | 45.4 | **1000.0** | **1000.0** | 559.0 | **1000.0** | 91.8 |
| Cartpole Swingup | 195.2 | 715.6 | **864.3** | 527.7 | 836.9 | 982.2 |
| Cartpole Swingup Sparse | 0.0 | 5.0 | 6.1 | 24.8 | **403.8** | 880.7 |
| Cheetah Run | 10.0 | 31.3 | 398.3 | 283.2 | **548.7** | 36.7 |
| Cup Catch | 0.0 | 826.7 | **979.2** | 729.6 | 973.3 | 910.9 |
| Finger Spin | 0.4 | 322.0 | **972.8** | 765.8 | 447.3 | 759.2 |
| Finger Turn Easy | 40.6 | 91.8 | 539.5 | 394.8 | **714.5** | 998.6 |
| Finger Turn Hard | 0.0 | 3.0 | 252.7 | 217.8 | **496.8** | 1000.0 |
| Hopper Hop | 0.0 | 1.0 | 0.2 | 0.0 | **51.6** | 282.7 |
| Hopper Stand | 4.5 | 16.7 | 4.4 | 58.7 | **199.0** | 347.6 |
| Pendulum Swingup | 0.0 | 371.0 | 744.3 | 830.4 | **875.0** | 980.6 |
| Reacher Easy | 38.6 | 527.3 | 936.6 | 693.4 | **943.8** | 841.1 |
| Reacher Hard | 28.0 | 95.7 | 591.2 | 768.0 | **878.8** | 881.0 |
| Walker Run | 25.3 | 126.8 | **756.9** | 243.2 | 311.1 | 680.3 |
| Walker Stand | 130.5 | 401.4 | **979.7** | 767.3 | 930.9 | 458.9 |
| Walker Walk | 37.8 | 390.9 | **971.7** | 475.2 | 876.7 | 446.7 |
| Mean | 50.0 | 331.7 | 619.1 | 456.1 | **647.4** | 650.7 |
| Median | 17.6 | 224.4 | 750.6 | 501.5 | **775.7** | 800.2 |

automated gradient-based method overwhelmingly outperforms this manual categorization, which barely improves upon the DreamerV3 baseline and even degrades the median performance.

This significant performance gap exists because manual categorization is inherently restrictive and suboptimal. A closer inspection reveals that our automated grouping actively diverges from human intuition to capture true optimization geometry. For example, based solely on semantic naming, one naturally groups tasks sharing the same embodiment together (such as Cartpole Balance, Cartpole Swingup, Cartpole Balance Sparse, and Cartpole Swingup Sparse). Conversely, human intuition almost certainly never groups Acrobot Swingup and Cup Catch into the same category. Yet, our method bypasses these surface-level constraints to cluster tasks based on actual gradient alignment, capturing subtle dynamic synergies that intuitive manual grouping completely misses.

This advantage becomes even more critical as task complexity scales. For instance, in the Meta-World MT50 benchmark, human intuition almost certainly groups all "Push" tasks together. However, our method automatically assigns "Push" and "Push Wall" to Group 1, while placing "Push Back" into Group 4. This data-driven separation highlights that the underlying dynamics and required feature representations for "Push Back" actually differ significantly from other pushing tasks, representing a critical nuance that manual categorization fails to recognize.

In summary, rather than relying on the prohibitive cost and inherent biases of human engineering, our method provides a principled, fully automated, and highly effective pipeline. It ensures robust performance and scalability to massive task suites without requiring any prior domain knowledge.

### D.4. Model Capacity

The introduction of the MoE module, which meaningfully integrates dynamics predictors from other groups, inherently increases our overall model size. To demonstrate that this capacity difference does not materially contribute to the performance gap, we conduct additional experiments on the Proprio Control benchmark by scaling up the Shared-Dreamer baseline. Specifically, we introduce two new baselines: Shared-Dreamer(big) (which matches our per-group model capacity) and Shared-Dreamer(large) (which matches the total parameter count of our method across all 9 groups). Following the standard scaling protocol of DreamerV3, we construct these larger baselines by proportionally increasing the base model dimension while keeping the network architecture and all hyperparameters strictly fixed. The experimental learning curves are shown in Figure 8, demonstrating that simply increasing the capacity of the shared model fails to provide consistent

*Table 11.* Full performance on the Visual Control benchmark demonstrates that TaskLoom can effectively learn from visual inputs. The best result in each row is highlighted in **bold**, while the second-best result is underlined.

| Task | PPO | CURL | DrQ-V2 | TD-MPC2 | DreamerV3 | TaskLoom (Ours) |
|---|---|---|---|---|---|---|
| Acrobot Swingup | 0.2 | 2.9 | 4.8 | 24.5 | 26.0 | **87.6** |
| Cartpole Balance | 367.8 | 982.5 | 987.4 | **989.7** | 935.9 | 988.5 |
| Cartpole Balance Sparse | 62.6 | **1000.0** | **1000.0** | 997.6 | 991.4 | **1000.0** |
| Cartpole Swingup | 56.6 | 551.2 | **626.4** | 400.4 | 598.5 | 277.3 |
| Cartpole Swingup Sparse | 0.0 | 0.0 | 22.4 | **83.6** | 56.7 | 12.8 |
| Cheetah Run | 7.3 | 259.6 | 273.2 | 243.4 | 299.5 | **552.0** |
| Cup Catch | 0.0 | 804.8 | 359.0 | 833.0 | 850.3 | **912.5** |
| Finger Spin | 0.2 | 797.6 | 325.2 | **980.9** | 502.1 | 186.0 |
| Finger Turn Easy | 99.4 | 287.2 | 161.3 | 341.3 | 408.7 | **499.5** |
| Finger Turn Hard | 0.0 | 197.9 | 99.9 | 298.7 | 68.9 | **515.0** |
| Hopper Hop | 0.4 | 5.3 | 3.0 | 38.9 | 8.1 | **56.2** |
| Hopper Stand | 0.0 | 126.7 | 20.1 | 62.4 | 275.7 | **459.0** |
| Pendulum Swingup | 0.0 | 38.0 | 234.6 | 380.6 | 296.0 | **462.5** |
| Quadruped Run | 105.3 | 106.3 | 154.5 | 68.8 | 111.3 | **240.6** |
| Quadruped Walk | 93.5 | 68.5 | 150.1 | 36.7 | 127.0 | **210.7** |
| Reacher Easy | 111.4 | 297.2 | 249.8 | 329.8 | 313.1 | **733.8** |
| Reacher Hard | 5.6 | 58.2 | 85.5 | 101.2 | 53.8 | **466.0** |
| Walker Run | 24.8 | 161.6 | 103.7 | **224.1** | 216.3 | 185.0 |
| Walker Stand | 127.3 | 720.7 | 324.4 | **934.7** | 900.5 | 712.3 |
| Walker Walk | 42.4 | 516.4 | 169.6 | 563.9 | **600.8** | 549.5 |
| Mean | 55.2 | 349.1 | 267.7 | 396.7 | 382.0 | **455.3** |
| Median | 16.0 | 228.7 | 165.4 | 314.2 | 297.8 | **464.3** |

*Table 12.* Success rate comparison on the Meta-World MT10 benchmark. Our method significantly outperforms both single-task and naive-sharing baselines, demonstrating its effectiveness in complex robotics domains.

| Algorithm | Success Rate |
|---|---|
| DreamerV3 | $0.300 \pm 0.129$ |
| Shared-Dreamer | $0.264 \pm 0.212$ |
| TaskLoom (Ours) | $\mathbf{0.517 \pm 0.075}$ |

improvements.

### D.5. Number of Groups

To verify the robustness of our method with respect to the number of groups, we conduct experiments on the Proprio Control benchmark. As shown in Table 17, all group sizes outperform the baseline. A group size of 9 (half of the 18 tasks) seems optimal in this setting, likely encouraging a balanced pairwise equilibrium that only breaks for strong gradient alignments, naturally filtering weak correlations. While this remains a hypothesis, the results confirm that our method is robust to the choice of the number of groups and mitigates negative transfer regardless of the specific hyperparameter chosen.

### D.6. Generalizability Across Model Architectures

To demonstrate the generalizability of our method across different formats of world models, we integrate our framework with TD-MPC2. We evaluate this new variant on the Proprio Control benchmark. To ensure a comprehensive comparison, we include two baselines: the standard online TD-MPC2 (where a separate model is trained independently for each task) and a Shared-TD-MPC2 (which simply forces all tasks to share a single, unified online world model).

As illustrated in Table 18, the Shared-TD-MPC2 approach suffers a severe performance drop compared to the single-task baseline. This highlights that naively mixing multi-task data is ineffective and triggers severe negative transfer, even in implicit decoder-free architectures. Conversely, applying our framework to TD-MPC2 safely and effectively leverages cross-task data, leading to clear performance improvements over the single-task baseline.

---

**Algorithm 1** TaskLoom Training Pipeline

---

1: **Input:** Total steps $T_{max}$, Stage 2 start step $T_{stage2}$.
2: **Initialize:** Task-specific replay buffers $\mathcal{D}_k \leftarrow \emptyset$ for each task $k \in \{1, \ldots, K\}$.
3: **Randomly initialize:** Group-specific world models $WM_g$ (including encoder, sequence model, dynamics $Dyn_g$, predictors, and decoder) for each group $g \in \{g_1, \ldots, g_N\}$.
4: **Randomly initialize:** Task-specific Actor-Critic networks $\{(\pi_{\theta_k}, V_{\psi_k})\}_{k=1}^K$.
5: **Define learning objective from RSSM backbone:** World Model Loss $\mathcal{L}(\phi)$, Actor Loss $\mathcal{L}(\theta)$, Critic Loss $\mathcal{L}(\psi)$.
6: *// Online Multi-Task Training Loop*
7: **for** environment step $t = 0$ **to** $T_{max}$ **do**
8:     *— Step 1: Continual Environment Interaction —*
9:     **for** $k = 1$ **to** $K$ **do**
10:         Collect transition using current $\pi_{\theta_k}$ and add to corresponding replay buffer $\mathcal{D}_k$.
11:     **end for**
12:     *— Step 2: Collaborative World Model Training & Policy Optimization —*
13:     **for each** group $g \in \{g_1, \ldots, g_N\}$ **do**
14:         Construct unified training batch from replay buffers: $X^{(t)} = \bigcup_{k \in g} X_k^{(t)}$.
15:         **if** $t < T_{stage2}$ **then**
16:             *(Stage 1: Intra-group)*
17:             Update group-specific world model $WM_g$ by minimizing the loss $\mathcal{L}(\phi)$ using $X^{(t)}$.
18:             Generate imagined rollouts in latent space using the updated $WM_g$.
19:             Update actor and critic by minimizing $\mathcal{L}(\theta)$ and $\mathcal{L}(\psi)$ using imagined rollouts.
20:         **else**
21:             *(Stage 2: Inter-group)*
22:             Freeze expert dynamics predictors $Dyn_i$ for all $i \neq g$.
23:             Compute $SK(h_t) = \sum_{i \neq g} \text{Softmax}(\text{Router}(h_t))_i Dyn_i(h_t)$.
24:             Substitute the prior $p(\hat{z}_t \mid Dyn_g(h_t))$ with $p(\hat{z}_t \mid Dyn_g(h_t) + SK(h_t))$ in $\mathcal{L}(\phi)$.
25:             Update $WM_g$ and SK Router by minimizing the modified loss $\mathcal{L}(\phi)$ using $X^{(t)}$.
26:             Generate imagined rollouts in latent space using the updated $WM_g$.
27:             Update actor and critic by minimizing $\mathcal{L}(\theta)$ and $\mathcal{L}(\psi)$ using imagined rollouts.
28:         **end if**
29:     **end for**
30: **end for**

---

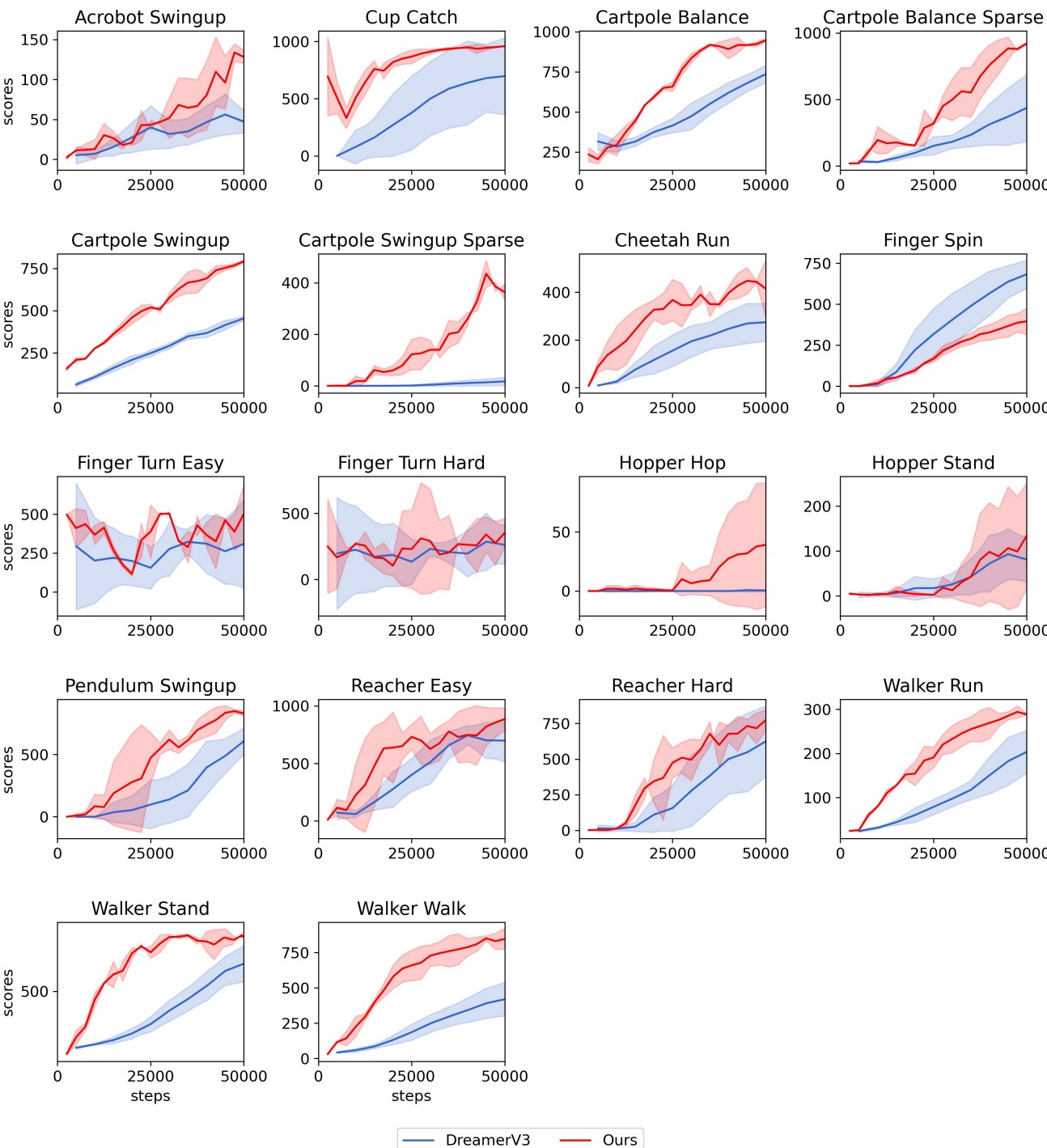

*Figure 4.* Training curves on the Proprio Control benchmark. Results of our method are depicted in red, whereas those of DreamerV3 (Hafner et al., 2025) are depicted in blue.

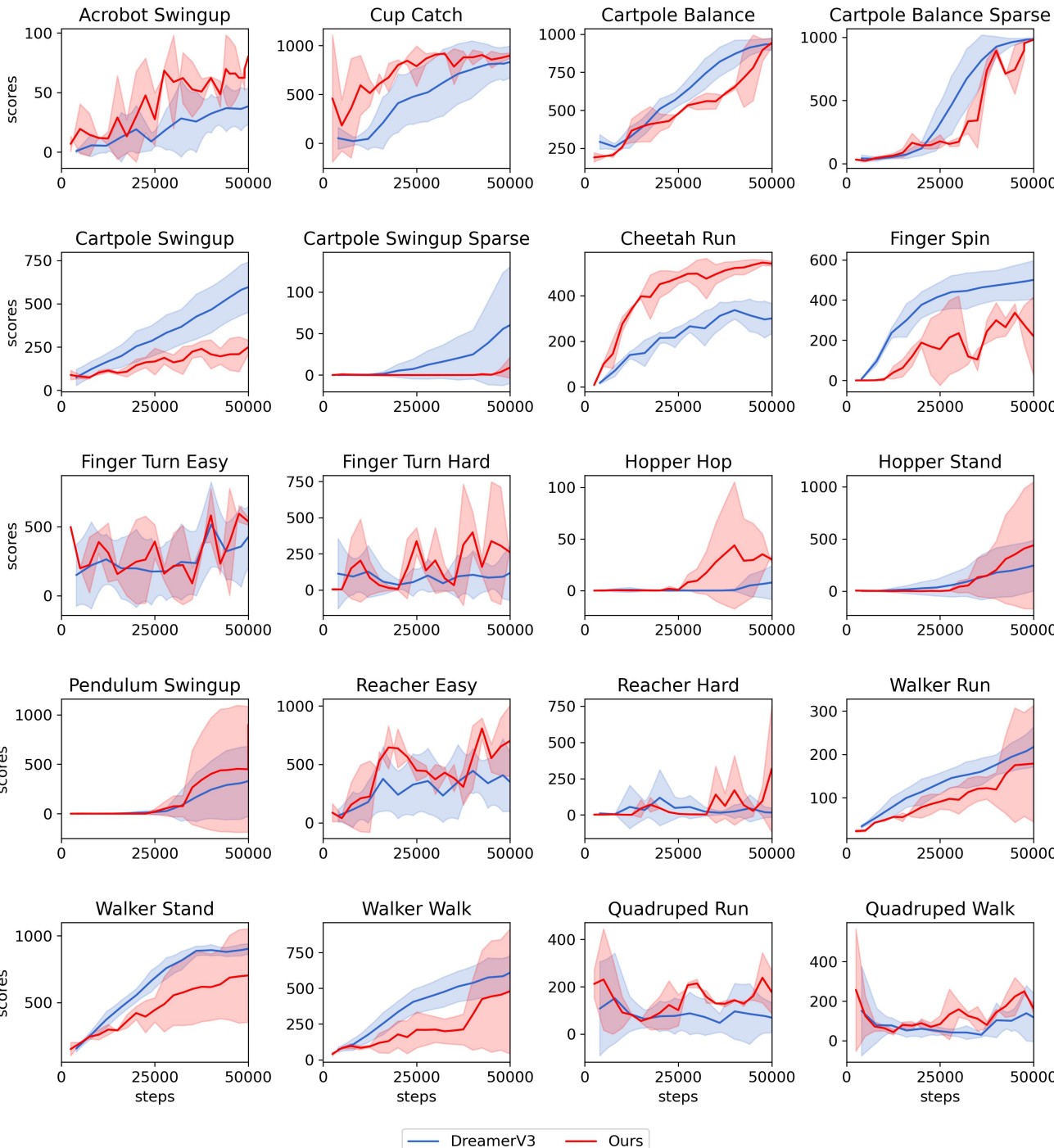

*Figure 5.* Training curves on the Visual Control benchmark. Results of our method are depicted in red, whereas those of DreamerV3 (Hafner et al., 2025) are depicted in blue.

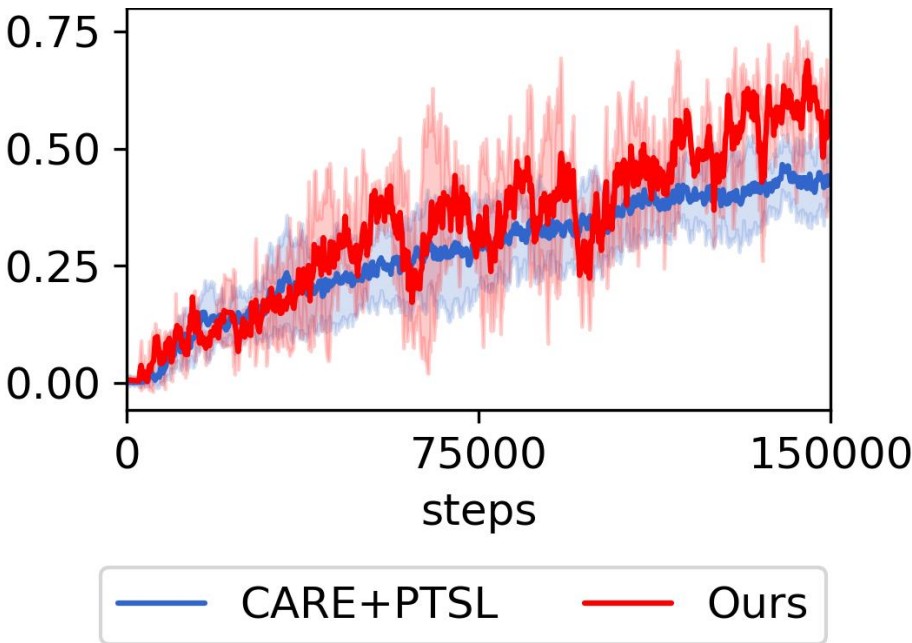

*Figure 6.* Training curves of our method compared with the strong baseline CARE+PTSL (Roberts & Di, 2024) on the MT10 benchmark. Results of our method are depicted in red, whereas those of CARE+PTSL are depicted in blue.

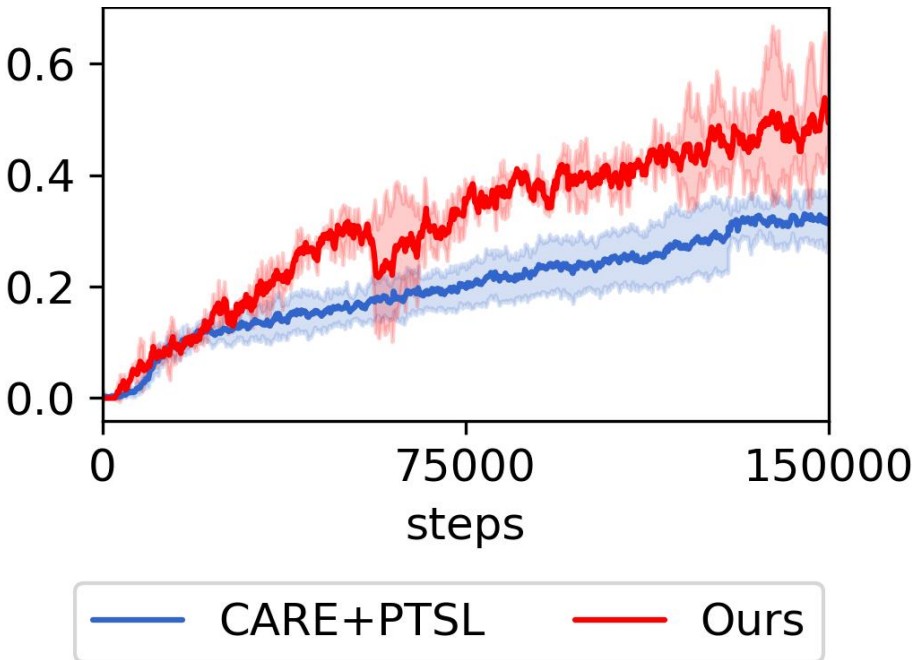

*Figure 7.* Training curves of our method compared with the strong baseline CARE+PTSL (Roberts & Di, 2024) on the MT50 benchmark. Results of our method are depicted in red, whereas those of CARE+PTSL are depicted in blue.

*Table 13.* Full ablation results on the Proprio Control benchmark.

| Task | DreamerV3 | Baseline | G-Baseline | G-Baseline+SK (Ours) | G-Baseline+SK-Dyn (Replacement) |
|---|---|---|---|---|---|
| Acrobot Swingup | 30.5 | 8.0 | 101.6 | **173.0** | 56.8 |
| Cartpole Balance | 839.7 | 466.1 | 966.4 | **992.9** | 989.5 |
| Cartpole Balance Sparse | 559.0 | 109.8 | 987.8 | **1000.0** | **1000.0** |
| Cartpole Swingup | 527.7 | 261.1 | 777.9 | **836.9** | 810.4 |
| Cartpole Swingup Sparse | 24.8 | 2.5 | 259.8 | 403.8 | **546.5** |
| Cheetah Run | 283.2 | 253.0 | 375.2 | **548.7** | 513.8 |
| Cup Catch | 729.6 | 503.8 | 471.5 | **973.3** | 863.0 |
| Finger Spin | **765.8** | 216.8 | 384.0 | 447.3 | 430.5 |
| Finger Turn Easy | 394.8 | 678.0 | 712.0 | **714.5** | 599.0 |
| Finger Turn Hard | 217.8 | 51.8 | 462.8 | **496.8** | 294.8 |
| Hopper Hop | 0.0 | 3.9 | 0.6 | **51.6** | 0.0 |
| Hopper Stand | 58.7 | 70.3 | 120.5 | **199.0** | 101.1 |
| Pendulum Swingup | 830.4 | 422.3 | 817.0 | **875.0** | 832.5 |
| Reacher Easy | 693.4 | 550.8 | 896.5 | **943.8** | 925.5 |
| Reacher Hard | 768.0 | 295.3 | 595.3 | **878.8** | 845.3 |
| Walker Run | 243.2 | 144.5 | 310.5 | 311.1 | **328.0** |
| Walker Stand | 767.3 | 703.6 | 923.6 | **930.9** | 880.1 |
| Walker Walk | 475.2 | 304.6 | 860.2 | **876.7** | 864.8 |
| Mean | 456.1 | 280.3 | 556.8 | **647.4** | 604.5 |
| Median | 501.5 | 257.0 | 533.4 | **775.7** | 704.7 |

*Table 14.* Performance implications of scaling the total number of training tasks. We report the performance on a fixed subset of 6 base tasks when the model is trained alongside 0, 6, or 12 additional tasks (totaling 6, 12, and 18 tasks, respectively). The best performance for each task is highlighted in **bold**.

| Task | 6 Tasks | 12 Tasks | 18 Tasks |
|---|---|---|---|
| Acrobot Swingup | 66.1 | 111.1 | **173.0** |
| Cup Catch | **973.5** | 949.5 | 973.3 |
| Finger Spin | 442.3 | 429.3 | **447.3** |
| Finger Turn Easy | 697.0 | 712.0 | **714.5** |
| Finger Turn Hard | 472.0 | 470.0 | **496.8** |
| Pendulum Swingup | 869.5 | 874.5 | **875.0** |
| Mean | 586.7 | 591.1 | **613.3** |
| Median | 584.5 | 591.0 | **605.6** |

*Table 15.* Performance comparison under an extended training horizon (250K steps) on the Proprio Control benchmark. The results demonstrate that our method maintains its clear advantage and achieves superior asymptotic performance over longer training runs. The best performance for each task is highlighted in **bold**.

| Task | PPO | DMPO | TD-MPC2 | DreamerV3 | TaskLoom (Ours) |
|---|---|---|---|---|---|
| Acrobot Swingup | 2.0 | 118.2 | 308.9 | 131.7 | **405.3** |
| Cartpole Balance | 551.1 | 998.3 | 996.0 | 990.6 | **998.6** |
| Cartpole Balance Sparse | 930.6 | 995.8 | **1000.0** | 1000.0 | 1000.0 |
| Cartpole Swingup | 265.8 | 859.6 | 866.7 | 856.0 | **878.5** |
| Cartpole Swingup Sparse | 1.8 | 454.5 | 775.3 | 503.6 | **843.5** |
| Cheetah Run | 100.5 | 603.4 | 769.9 | 596.3 | **829.4** |
| Cup Catch | 503.2 | 970.4 | **978.7** | 966.0 | 968.3 |
| Finger Spin | 17.8 | 780.6 | **985.3** | 924.4 | 657.5 |
| Finger Turn Easy | 266.8 | 482.0 | 884.7 | 793.0 | **988.8** |
| Finger Turn Hard | 392.8 | 260.8 | 752.3 | 932.8 | **980.3** |
| Hopper Hop | 0.0 | 64.2 | **208.4** | 107.5 | 196.7 |
| Hopper Stand | 0.0 | 545.3 | **873.3** | 650.4 | 862.6 |
| Pendulum Swingup | 0.0 | 819.0 | 638.9 | 785.0 | **877.8** |
| Reacher Easy | 662.2 | 977.7 | 914.6 | 961.8 | **983.0** |
| Reacher Hard | 168.4 | 965.3 | 870.2 | 948.6 | **977.0** |
| Walker Run | 31.4 | 511.5 | **788.6** | 655.5 | 391.5 |
| Walker Stand | 151.7 | 980.2 | 989.7 | 957.2 | **993.0** |
| Walker Walk | 77.0 | 938.9 | **976.8** | 881.0 | 964.0 |
| Mean | 229.1 | 684.8 | 809.9 | 757.8 | **822.0** |
| Median | 126.1 | 799.8 | 871.7 | 868.5 | **921.3** |

*Table 16.* Performance comparison between our automated gradient-based grouping and intuitive manual grouping on the Proprio Control benchmark. The automated approach significantly outperforms manual categorization, highlighting the limitations of relying on human intuition for multi-task optimization. The best performance for each task is highlighted in **bold**.

| Task | DreamerV3 | Intuitive Manual Grouping | Ours (Gradient-Based Grouping) |
|---|---|---|---|
| Acrobot Swingup | 30.5 | 86.1 | **173.0** |
| Cartpole Balance | 839.7 | 883.0 | **992.9** |
| Cartpole Balance Sparse | 559.0 | 259.0 | **1000.0** |
| Cartpole Swingup | 527.7 | 638.2 | **836.9** |
| Cartpole Swingup Sparse | 24.8 | 27.8 | **403.8** |
| Cheetah Run | 283.2 | 281.5 | **548.7** |
| Cup Catch | 729.6 | 958.0 | **973.3** |
| Finger Spin | **765.8** | 320.0 | 447.3 |
| Finger Turn Easy | 394.8 | 500.0 | **714.5** |
| Finger Turn Hard | 217.8 | **601.5** | 496.8 |
| Hopper Hop | 0.0 | **62.3** | 51.6 |
| Hopper Stand | 58.7 | 153.6 | **199.0** |
| Pendulum Swingup | 830.4 | 126.0 | **875.0** |
| Reacher Easy | 693.4 | **956.5** | 943.8 |
| Reacher Hard | 768.0 | 748.5 | **878.8** |
| Walker Run | 243.2 | 185.8 | **311.1** |
| Walker Stand | 767.3 | **938.8** | 930.9 |
| Walker Walk | 475.2 | **888.0** | 876.7 |
| Mean | 456.1 | 478.6 | **647.4** |
| Median | 501.5 | 410.0 | **775.7** |

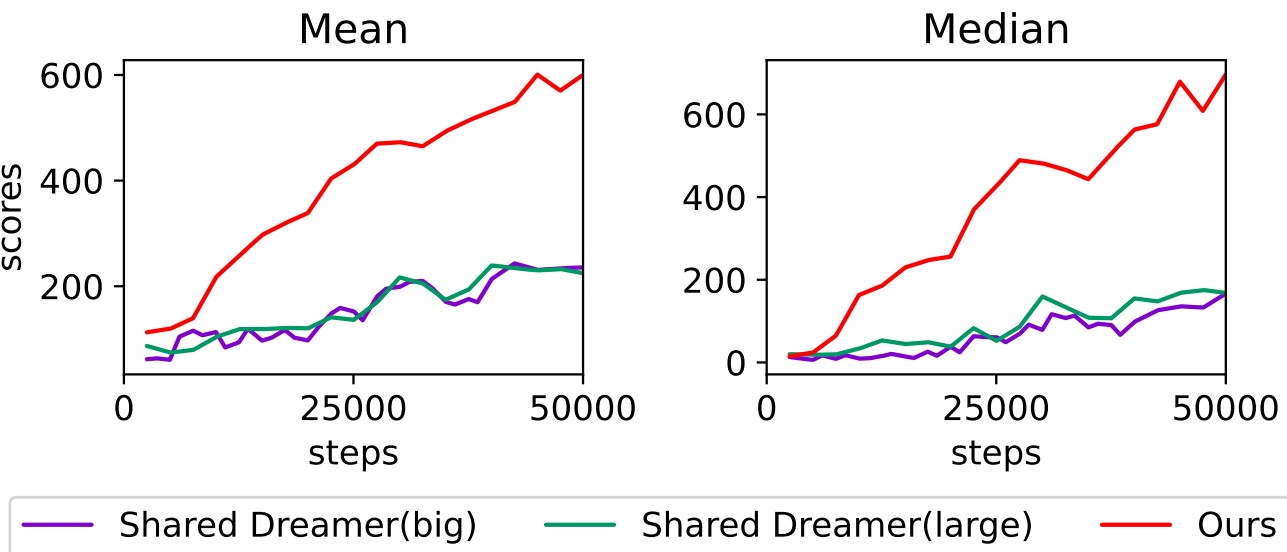

*Figure 8.* Performance comparison among Shared-Dreamer(big), Shared-Dreamer(large) and our method.

*Table 17.* Ablation study on the number of groups. We evaluate our method across different group sizes on the Proprio Control benchmark.

| Task | DreamerV3 | Group 3 | Group 6 | Group 9 (Half) | Group 12 | Group 15 |
|---|---|---|---|---|---|---|
| Acrobot Swingup | 30.5 | 68.4 | 107.8 | 173.0 | 204.1 | 345.0 |
| Cartpole Balance | 839.7 | 721.0 | 983.7 | 992.9 | 986.9 | 989.8 |
| Cartpole Balance Sparse | 559.0 | 495.0 | 722.5 | 1000.0 | 1000.0 | 994.5 |
| Cartpole Swingup | 527.7 | 427.5 | 553.2 | 836.9 | 812.0 | 839.0 |
| Cartpole Swingup Sparse | 24.8 | 16.3 | 11.8 | 403.8 | 718.3 | 0.0 |
| Cheetah Run | 283.2 | 435.4 | 531.7 | 548.7 | 522.7 | 519.4 |
| Cup Catch | 729.6 | 970.8 | 952.5 | 973.3 | 970.3 | 943.5 |
| Finger Spin | 765.8 | 419.0 | 406.8 | 447.3 | 417.5 | 487.0 |
| Finger Turn Easy | 394.8 | 739.0 | 756.0 | 714.5 | 500.0 | 480.0 |
| Finger Turn Hard | 217.8 | 737.0 | 728.5 | 496.8 | 485.8 | 698.3 |
| Hopper Hop | 0.0 | 70.3 | 50.2 | 51.6 | 21.5 | 35.8 |
| Hopper Stand | 58.7 | 234.5 | 141.4 | 199.0 | 176.6 | 232.8 |
| Pendulum Swingup | 830.4 | 846.5 | 859.8 | 875.0 | 447.3 | 936.8 |
| Reacher Easy | 693.4 | 975.8 | 883.3 | 943.8 | 932.8 | 560.8 |
| Reacher Hard | 768.0 | 921.8 | 920.5 | 878.8 | 744.5 | 942.3 |
| Walker Run | 243.2 | 313.5 | 289.9 | 311.1 | 286.3 | 317.4 |
| Walker Stand | 767.3 | 982.4 | 967.4 | 930.9 | 927.9 | 952.5 |
| Walker Walk | 475.2 | 849.9 | 918.5 | 876.7 | 924.7 | 765.1 |
| Mean | 456.1 | 568.0 | 599.2 | 647.4 | 615.5 | 613.3 |
| Median | 501.5 | 608.0 | 725.5 | 775.7 | 620.5 | 629.5 |

*Table 18.* Performance comparison on the Proprio Control benchmark. We integrate our framework with TD-MPC2 to demonstrate its generalizability. For each task, the best-performing method is highlighted in **bold**, and the second-best is underlined.

| Task | TD-MPC2 | Shared-TD-MPC2 | Ours (TD-MPC2 version) |
|------|---------|----------------|------------------------|
| Acrobot Swingup | **182.7** | 9.0 | 33.3 |
| Cartpole Balance | **964.2** | 532.1 | 892.5 |
| Cartpole Balance Sparse | **1000.0** | 140.0 | **1000.0** |
| Cartpole Swingup | **864.3** | 402.8 | 764.2 |
| Cartpole Swingup Sparse | 6.1 | 42.5 | **777.7** |
| Cheetah Run | **398.3** | 59.0 | 284.7 |
| Cup Catch | **979.2** | 599.3 | 954.7 |
| Finger Spin | **972.8** | 2.2 | 289.5 |
| Finger Turn Easy | 539.5 | 258.3 | **832.3** |
| Finger Turn Hard | 252.7 | 166.8 | **604.1** |
| Hopper Hop | 0.2 | 0.3 | **7.3** |
| Hopper Stand | 4.4 | 6.0 | **44.4** |
| Pendulum Swingup | 744.3 | 3.5 | **790.1** |
| Reacher Easy | **936.6** | 385.0 | 917.9 |
| Reacher Hard | 591.2 | 71.5 | **872.5** |
| Walker Run | **756.9** | 165.3 | 500.3 |
| Walker Stand | 979.7 | 628.9 | **990.4** |
| Walker Walk | **971.7** | 500.7 | 953.3 |
| Mean | 619.1 | 220.7 | **639.4** |
| Median | 750.6 | 152.7 | **783.9** |

