# OpenReview forum: "TaskLoom: Weaving Knowledge Across Tasks in World Models"
_ICML.cc/2026/Conference — ICML 2026 regular_

### Official Review · Reviewer_SZbo · 2026-03-08

**Soundness:** 3
**Presentation:** 2
**Significance:** 3
**Originality:** 3
**Overall Recommendation:** 4
**Confidence:** 3

**Summary:**

The article aims to explore a notable aspect of online multi-task model-based reinforcement learning: how to share environment dynamics across tasks through a structured world model rather than learning separate models for each task. Overall, a critical issue studied is how to leverage shared structure among tasks while avoiding negative transfer that often arises in naive multi-task training.
The paper proposes TaskLoom, a grouped two-stage training framework built on RSSM world models (e.g., DreamerV3). First, tasks are clustered according to gradient similarity to form groups, and tasks within each group share a world model to enable fine-grained knowledge transfer. Second, a shared knowledge module based on a mixture-of-experts architecture allows coarse-grained dynamics knowledge to be exchanged across groups through a residual correction to the group-specific dynamics predictor. Experiments on DMControl (both proprioceptive and visual control) and Meta-World benchmarks show consistent improvements over DreamerV3, TD-MPC2, and several multi-task RL baselines.

**Compliance With Llm Reviewing Policy:**

Affirmed.

**Final Justification:**

The authors addressed my questions, so I will keep the score.

**Key Questions For Authors:**

1. What does $M$ stand for? It is not defined in the paper. Do you do gradient clustering for M times?

2. When describing the RSSM that your paper is built on, can you say a bit more on why deterministic hidden states along with stochastic
latent variables helps and what kinds of things each of them is learning? Also, how does $\hat z_t$ (dynamic predicted stochastic latent variable) play a role? What are each of the three terms in the loss?

**Limitations:**

First, the task grouping procedure is largely heuristic and lacks theoretical justification. The number of groups is fixed a priori, and the sensitivity of the method to this choice is not systematically studied. Incorrect grouping could potentially lead to negative transfer.
Second, the shared knowledge mechanism does not provide clear interpretability or analysis of what information is actually transferred across groups. The mixture-of-experts module is introduced empirically, and the paper does not provide insight into how shared dynamics knowledge is represented.
Third, the framework only shares world models while leaving actor–critic networks task-specific. This limits the method’s applicability to broader multi-task reinforcement learning settings where policies must generalize across tasks.

**Strengths And Weaknesses:**

Soundness: Strong
The proposed approach is technically sound and well grounded in existing world model frameworks. The method builds on Dreamer-style RSSM architectures and introduces clearly defined modifications, including gradient-based task grouping and a mixture-of-experts residual dynamics module. The experimental protocol is thorough, covering both low-dimensional and visual control tasks, and includes extensive ablation studies that isolate the contributions of task grouping and the shared knowledge module. These experiments support the claims made in the paper and demonstrate consistent performance improvements.

Significance: Moderate
The empirical results are strong. However, the conceptual impact is somewhat limited. The main contributions lie in combining existing techniques—task clustering, mixture-of-experts routing, and world model sharing—rather than introducing fundamentally new algorithmic principles. Additionally, the approach keeps actor–critic networks task-specific, meaning the framework does not fully address multi-task policy learning.

Presentation: Moderate
The paper is generally well written. The motivation for sharing world model knowledge across tasks is well articulated, and the architecture of TaskLoom is explained with helpful diagrams. However, the introduction section falls somewhat underdeveloped. For instance, the observation notation seems to change from $o$ POMDP to $z$ and $h$ later.

Originality: Moderate
The paper’s novelty primarily lies in the integration of existing ideas rather than the introduction of new theoretical or algorithmic insights. Task grouping based on gradient similarity, hierarchical sharing structures, and mixture-of-experts architectures have all been explored in related literature. TaskLoom’s contribution is mainly the application and combination of these ideas within the context of world model learning for online multi-task RL. While this integration is effective, the level of originality appears moderate.

---

> ### Author Rebuttal · Authors · 2026-03-31
>
> **Question 1**
>
> Apologies for the omission. Yes, we do gradient clustering for $M$ times. We will explicitly define this in the revision.
>
> **Question 2**
>
> We thank the reviewer for this insightful question. While RSSM (from DreamerV3) is our backbone rather than our core innovation, we will add this summary:
> * **Deterministic & Stochastic States:** The deterministic hidden state $h_t$ acts as memory, capturing long-term dependencies to resolve partial observability. The stochastic latent variable $z_t$, inferred from current observations, models environmental uncertainty. Together, they capture distinct possible futures, avoiding the "blurry" predictions typical of purely deterministic models.
> * **Role of $\hat{z}_t$:** $\hat{z}_t$ is the dynamics predictor's output (the "prior"). It predicts the next stochastic state using only historical context $h_t$. This enables latent imagination: recursively predicting $\hat{z}_t$ allows the actor-critic to learn sample-efficient policies entirely in latent space.
> * **Losses (sg = stop-gradient):**
>   Prediction: $\mathcal{L}\_{pred}(\phi) = - \ln p\_\phi(\hat{o}\_t | h\_t, z\_t) - \ln p\_\phi(\hat{r}\_t | h\_t, z\_t) - \ln p\_\phi(\hat{c}\_t | h\_t, z\_t)$
>
>   Dynamics: $\mathcal{L}\_{dyn}(\phi) = \max \Big( 1, \operatorname{KL} \big[ \operatorname{sg}(q\_\phi(z\_t | h\_t, o\_t)) | p(\hat{z}\_t | Dyn\_\phi(h\_t)) \big] \Big)$
>
>   Representation: $\mathcal{L}\_{rep}(\phi) = \max \Big( 1, \operatorname{KL} \big[ q\_\phi(z\_t | h\_t, o\_t) | \operatorname{sg}(p(\hat{z}\_t | Dyn\_\phi(h\_t))) \big] \Big)$
> In Stage 2 of TaskLoom, we upgrade the prior with Shared Knowledge ($SK$): $\hat{z}\_t \sim p(\hat{z}\_t | Dyn\_g(h\_t) + SK(h\_t))$. This leverages $\mathcal{L}\_{dyn}$ to pull predictions toward globally plausible outcomes and forces $\mathcal{L}\_{rep}$ to accommodate shared knowledge.
>
> **Notation Clarity**
>
> Apologies for the confusion. As in Sec 3.2, $o$ is observation, $z$ is stochastic latent, and $h$ is deterministic hidden state. We will standardize this.
>
> **Limitation 1**
>
> Our grouping is driven by empirical gradient similarity, capturing true optimization geometry. Setting group counts a priori is standard in multi-task learning frameworks using K-Means (e.g., M3DT), but our method is robust to it. For example, on Proprio Control (18 tasks), all group sizes outperform baseline:
> | Metric | DreamerV3 (baseline) | Groups 3 | Groups 6 | Groups 9 (Half) | Groups 12 | Groups 15 |
> | :--- | :--- | :--- | :--- | :--- | :--- | :--- |
> | Mean | 456.1 | 568.0 | 599.2 | 647.4 | 615.5 | 613.3 |
> | Median | 501.5 | 608.0 | 725.5 | 775.7 | 620.5 | 629.5 |
>
> Groups=9 (half of 18) seems optimal in this setting, likely encouraging a balanced pairwise equilibrium that only breaks for strong gradient alignments, naturally filtering weak correlations. While this remains a hypothesis, the results confirm that our method is robust to the choice of number of groups and mitigates negative transfer regardless of the specific hyperparameter chosen.
>
> **Limitation 2**
>
> Our MoE is fundamentally designed to selectively filter and integrate generalizable knowledge across task groups. Because it operates on the dynamics predictors, the transferred information is the world model's "Prior" (abstract transition logic and physics) (See also Question 2).
>
> Crucially, this shared knowledge is explicitly represented and quantified by the learned routing weights. The router dynamically assigns larger weights to leverage complementary, highly applicable knowledge from correlated groups. Conversely, it assigns smaller weights to filter out redundant or irrelevant dynamics that would otherwise introduce negative transfer.
>
> To empirically validate this, we visualized predictions by isolating specific experts (Figure: https://anonymous.4open.science/r/TaskLoom-Rebuttal/Figure.png). The model without MoE performs the worst. While relying solely on low-weight experts slightly improves upon this baseline, it still exhibits significant deviations. In contrast, using only high-weight experts maintains excellent performance, closely matching both the model with full MoE (Ours) and the Ground Truth. This ablation explicitly confirms that routing weights accurately quantify meaningful dynamic synergies.
>
> **Limitation 3**
>
> Though focused on online world model inefficiencies (typically single-task), TaskLoom is highly flexible and accommodates unified policy learning. Training a unified actor-critic on Proprio Control drops performance compared to task-specific actors, but still outperforms the baseline:
> | Metric | DreamerV3 | Ours (Unified actor for all tasks) | Ours (Task-specific actors) |
> | :--- | :--- | :--- | :--- |
> | Mean | 456.1 | 622.6 | 647.4 |
> | Median | 501.5 | 735.8 | 775.7 |
>
> This confirms that TaskLoom is not inherently limited to task-specific policies and holds strong potential for broader multi-task RL settings where a unified policy is desired.

---

> > ### Author Rebuttal · Reviewer_SZbo · 2026-04-01
> >
> > All questions have been answered, and I will keep the positive score.

---

> > > ### Author Response · Authors · 2026-04-02
> > >
> > > We sincerely thank the reviewer for evaluating our rebuttal and for the continued positive support. We are very glad that our responses have fully addressed your concerns, and we deeply appreciate your constructive feedback.

---

### Official Review · Reviewer_x6Dn · 2026-03-10

**Soundness:** 3
**Presentation:** 3
**Significance:** 3
**Originality:** 3
**Overall Recommendation:** 4
**Confidence:** 4

**Summary:**

This paper proposes TaskLoom, a knowledge-sharing world model for online RL. TaskLoom consists of two stages of training: 1) dividing tasks into several groups according to gradient similarities and training a model for those tasks within the same group; 2) then using a MoE model to incorporate cross-group information to further improve dynamics predictions. Experimental results show that TaskLoom outperforms baseline methods on widely used benchmarks such as Proprio Control, Visual Control and Meta-World.

**Compliance With Llm Reviewing Policy:**

Affirmed.

**Final Justification:**

After the authors' rebuttal, my concerns are fully resolved.

**Key Questions For Authors:**

1. The paper states that all methods are trained with 50k samples on the DMC benchmark. Is it 50k samples per task, or 50k samples in total across all tasks?
2. Could the authors evaluate the proposed method using a TD-MPC2-style world model architecture to verify whether the approach is effective beyond RSSM-based models?

**Limitations:**

No. I don't find any discussion of limitations.

**Strengths And Weaknesses:**

### Strengths

1. The experiments are very comprehensive. The paper evaluates the proposed algorithm across multiple benchmarks and includes detailed ablations to analyze the contribution of each component.
2. The algorithm design is intuitive, and the authors repeatedly draw analogies to human cognition throughout the paper, which helps readers understand the underlying philosophy behind each component.

### Weaknesses

1. Many of the conclusions in the paper appear to be validated only under the RSSM architecture. For example, the ablation study shows that training a single world model across multiple datasets performs significantly worse than training separate models for each task. However, it is unclear whether this conclusion is specific to RSSM-based models. The authors should verify whether the proposed method also works with other world model architectures, such as the model structure used in TD-MPC2.
2. The gradient-similarity-based task grouping method proposed by the authors appears somewhat complex. Based on the results, it seems that simple manual grouping might achieve similar outcomes. For instance, in Table 5 (Proprio Control benchmark), Reacher tasks are grouped together, Walker tasks are grouped together, and Hopper tasks form another group, which largely aligns with intuitive manual categorization.

---

> ### Author Rebuttal · Authors · 2026-03-31
>
> **Weakness 1 & Question 2**
>
> We appreciate the reviewer for raising this insightful question regarding the broader applicability of our method. To verify that our proposed framework generalizes beyond RSSM-based architectures, we integrated our approach with TD-MPC2.
>
> We evaluated this new variant on the Proprio Control benchmark. To ensure a comprehensive comparison, we included two baselines: the standard online **TD-MPC2** (where a separate model is trained independently for each task) and a **Shared-TD-MPC2** (which simply forces all tasks to share a single, unified online world model).
>
> As illustrated in the table below, the Shared-TD-MPC2 approach suffers a severe performance drop compared to the single-task baseline. This highlights that naively mixing multi-task data is ineffective and triggers severe negative transfer, even in implicit decoder-free architectures. Conversely, applying our framework to TD-MPC2 safely and effectively leverages cross-task data, leading to clear performance improvements over the single-task baseline.
>
> | Metric | TD-MPC2 | Shared-TD-MPC2 | Ours (TD-MPC2 version) |
> | :--- | :--- | :--- | :--- |
> | Mean | 619.1 | 220.7 | 639.4 |
> | Median | 750.6 | 152.7 | 783.9 |
>
> **Importantly, we integrated our framework with TD-MPC2 in a direct manner without conducting any architecture-specific hyperparameter tuning.** The fact that our method achieves these clear improvements under such a zero-tuning setting demonstrates that TaskLoom is not strictly limited to RSSM formats, but can be successfully and robustly generalized to other types of world models. We will add a discussion on this generalizability in the revised manuscript.
>
> **Weakness 2**
>
> We appreciate the reviewer's observation. However, we would like to emphasize that the core objective of our gradient-based mechanism is to provide an **automated, scalable, general, and highly effective** pipeline. While it may capture certain intuitive similarities in simpler benchmarks (like grouping Reacher tasks), its true value lies in extracting underlying dynamic synergies directly from data, completely removing the bottleneck of human bias and the need for prior domain knowledge.
>
> To directly address the hypothesis that simple manual categorization might achieve similar outcomes, we evaluated an **Intuitive Manual Grouping** baseline on the Proprio Control benchmark. In this baseline, tasks are grouped strictly according to their shared embodiment (e.g., placing all Cartpole tasks in one group and all Walker tasks in another, exactly as human intuition would dictate). As shown in the table below, our automated gradient-based method overwhelmingly outperforms this manual categorization, which barely improves upon the baseline and even degrades the median performance.
>
> | Metric | DreamerV3 (Baseline) | Intuitive Manual Grouping | Ours (Gradient-Based Grouping) |
> | :--- | :--- | :--- | :--- |
> | Mean | 456.1 | 478.6 | **647.4** |
> | Median | 501.5 | 410.0 | **775.7** |
>
> This severe performance gap exists because manual categorization is inherently restrictive and suboptimal. A closer inspection reveals that our automated grouping actively diverges from human intuition to capture true optimization geometry. For example, based solely on semantic naming, one would naturally group tasks sharing the same embodiment together, such as "Cartpole Balance", "Cartpole Swingup", "Cartpole Balance Sparse", and "Cartpole Swingup Sparse". Conversely, intuition would almost certainly never group "Acrobot Swingup" and "Cup Catch" into the same category. Yet, our method bypasses these surface-level constraints to cluster tasks based on actual gradient alignment, capturing subtle dynamic synergies that intuitive manual grouping completely misses.
>
> This advantage becomes even more critical as complexity scales. For instance, in the Meta-World MT50 benchmark, human intuition would almost certainly group all "Push" tasks together. However, our method automatically assigns "Push" and "Push Wall" to Group 1, while placing "Push Back" into Group 4. This data-driven separation highlights that the underlying dynamics and required feature representations for "Push Back" actually differ significantly from other pushing tasks, which is a critical nuance that manual categorization would fail to recognize.
>
> In summary, rather than relying on the prohibitive cost and inherent biases of human engineering, our method provides a principled, fully automated, and highly effective pipeline. It ensures robust performance and scalability to massive task suites without requiring any prior domain knowledge.
>
> **Question 1**
>
> To clarify, all methods are trained with **50k samples per task**. (See also Response to Reviewer q1qV: Weakness 2 & Question 2)

---

> > ### Author Rebuttal · Reviewer_x6Dn · 2026-04-01
> >
> > My concerns are fully resolved, and I will raise my score.

---

> > > ### Author Response · Authors · 2026-04-02
> > >
> > > We sincerely thank the reviewer for evaluating our rebuttal and for raising the score. We are very glad that our responses have fully resolved your concerns, and we deeply appreciate your constructive feedback.

---

### Official Review · Reviewer_q1qV · 2026-03-12

**Soundness:** 2
**Presentation:** 1
**Significance:** 2
**Originality:** 2
**Overall Recommendation:** 4
**Confidence:** 4

**Summary:**

The paper proposes a two-stage approach for training a multi-task world model built on DreamerV3-style RSSM dynamics. The core idea is to exploit similarities across tasks to reduce the total wall-clock training time. To achieve this, the authors use a grouped, two-stage training paradigm: in the first stage, they cluster tasks into groups based on the similarity of world-model gradients; in the second stage, they exchange coarse-grained knowledge across groups. Empirically, the approach shows strong results compared with the baselines.

**Compliance With Llm Reviewing Policy:**

Affirmed.

**Final Justification:**

I'm increasing my rating to weak accept.
The 2nd round rebuttal experiment results strengthen the paper. The authors extend the proprio control training to 250k (5x) and the proposed method performance continues to improve. The fairness in the comparison is also improved. The author reports the gradient update as well.

However, my weak acceptance is conditional on the authors committing to lifting all remaining unclear or underspecified parts in the final revision, specifically:
1. Define the step-counting convention precisely in the multi-task parallel setting (what exactly counts as an “environment step” when interacting with K tasks in parallel), and report the exact optimiser-step accounting for world model / actor&critic (per collected transition and/or per global training iteration).
2. Integrate the full training pseudocode into the main paper, and include the missing operational details that materially affect training/compute in appendix (update schedule, batch construction, and any warmup/regrouping procedure).
3. Include all the learning curves of the additional experiments done in the 2nd round rebuttal.

**Key Questions For Authors:**

1. What is the pseudocode of the training pipeline including the actor critic training. Are the world model and the policy jointly trained or separate?
2. How exactly the 50K steps map to your training pipeline? Is it fair to compare with other baselines?
3. How much memory required during training and evaluation?

**Limitations:**

As the weakness.

**Strengths And Weaknesses:**

Strengths:
1. The problem is clearly framing and "one world model per task is wasteful" is true given that there are many redundant dynamics.
2. Empirical ablation results support the design. Naively learning the world model dynamic results in a performance degeneration.

Weakness:
1.The actual training pipeline is too vague to understand the full picture. What is the exactly timeline of stage 1, stage 2 and the policy learning. Are they trained jointly? If yes, how? As the line 246 said "At step t, the training data from task k {1,2,...,k_group} ...". Do you need to train the actor critic in the same time as you train the multi-task world model? If actor–critic training is delayed or decoupled, how do the authors address potential distribution shift in the world model as the policy improves

2. 50k interaction steps in Table 1 is also vague. Do you mean 50k interaction per task or per group or in total? Without a clearly state, it's hard to understand if the comparison is fair.

3. How big is the world model? How much memory is required during training and evaluation? With the concurrent training and MoE setting for the world model, it seems that all the world models for different groups need to be in the memory during training and evaluation as you are using DreamerV3 style RSSM world model.

---

> ### Author Rebuttal · Authors · 2026-03-31
>
> **Weakness 1 & Question 1**
>
> We thank the reviewer for highlighting the need for a clearer training pipeline. To clarify, our world model and the policy (actor-critic) are **jointly trained**. Specifically, our pipeline interleaves the updates of the world model and the actor-critics within the same overarching training loop. Because the policies are updated using trajectories imagined by the most up-to-date world model, this tightly coupled scheme naturally mitigates any potential distribution shifts that might occur if policy learning were delayed or decoupled.
>
> To provide a comprehensive view of the full pipeline (including Stage 1, Stage 2, and the concurrent policy learning), we have provided the detailed pseudocode in an anonymous repository here: https://anonymous.4open.science/r/TaskLoom-Rebuttal/pseudocode.png. We will include this pseudocode in the revised manuscript.
>
> **Weakness 2 & Question 2**
>
> We apologize for the ambiguity. The "50k interaction steps" reported in Table 1 strictly refers to **50k steps per task**.
>
> This setting ensures a completely fair comparison, as both our method and the baselines are constrained to interact with each online task environment for exactly 50k steps. The key difference lies in data utilization. Traditional single-task world models discard interaction data from other tasks because lacking a principled way to share this information often leads to negative transfer. To overcome this, our method introduces a carefully designed hierarchical, group-based knowledge sharing mechanism to safely and effectively leverage cross-task data for improved performance.
>
> Furthermore, our ablation study using the "Shared-Dreamer" baseline demonstrates that simply forcing all tasks to share a single, unified world model actually leads to worse performance than training separate models for each task. This degradation highlights that naively mixing 50k steps of data across different tasks is ineffective. It also underscores the non-trivial nature of our contribution: our carefully designed framework successfully harnesses cross-task data where naive sharing fails, proving the significance of our approach. We will make the "50k steps per task" explicitly clear in the revised text.
>
> **Weakness 3 & Question 3**
>
> We appreciate the opportunity to clarify the model size and memory constraints. Contrary to the concern that our concurrent training and MoE setting might require excessive memory, our approach actually **reduces** both the total parameter count and the total memory footprint compared to training independent world models for the same number of tasks.
>
> By grouping tasks, we significantly decrease the total number of world models required (e.g., cutting them in half to 9 groups for 18 tasks). While the inclusion of the MoE module (which integrates dynamics predictors from other groups) increases the parameter size of a single group's world model from 12M to 21M, our total parameter count across all tasks remains strictly lower than the baseline (189M vs. 216M). Furthermore, during Stage 2, we do not need to load the complete world models of other groups into memory, ensuring memory efficiency.
>
> To provide a comprehensive view, we summarize the network parameters, memory footprint, and overall performance in the table below (using the Proprio Control benchmark with 18 tasks as the example).
>
> | Metric | DreamerV3 (Independent) | Shared-Dreamer | Ours |
> | :--- | :--- | :--- | :--- |
> | **World Model Parameters** | 216M (12M x 18) | 12M (12M x 1) | 189M (21M x 9) |
> | **Training Memory** | 28,800 MB (1,600 MB x 18) | 3,100 MB (3,100 MB x 1) | 27,000 MB (3,000 MB x 9) |
> | **Evaluation Memory**| 171 MB (9.5 MB x 18) | 10.9 MB (10.9 MB x 1) |90 MB (10 MB x 9) |
> | **Mean** | 456.1 | 280.3 | **647.4** |
> | **Median** | 501.5 | 257.0 | **775.7** |
>
> As demonstrated above, even at its peak training memory usage in Stage 2 (~3,000 MB per group), our method requires less total memory (27,000 MB) than maintaining separate standard world models (28,800 MB). Furthermore, during evaluation, our memory footprint is nearly cut in half compared to the independent baseline. Most importantly, our framework achieves this superior computational efficiency while delivering significantly higher overall performance and avoiding the severe negative transfer seen in the naive Shared-Dreamer approach.

---

> > ### Author Rebuttal · Reviewer_q1qV · 2026-04-02
> >
> > 1. Could you report results under a longer interaction budget as a reference? The current experiments leave a potential confound: the observed gains may reflect faster convergence rather than improved final/asymptotic performance (and could even come at the cost of worse long-run performance). I understand that running the full 20×5 DMControl suite to 1M steps is computationally expensive, but reporting longer-horizon results on a representative subset of tasks would significantly strengthen the paper.
> > 2. Could you report the actual number of gradient updates implied by your training loop—either directly in the pseudocode or as a Dreamer-style update-to-data ratio in the hyperparameters? As written, the paper does not specify how many world model / actor critic gradient steps are performed per environment interaction step, so the training compute behind “50K steps” is unclear.
> > 3. Model capacity fairness: Shared-Dreamer uses a ~12M-parameter world model, whereas your Stage-2 group model is ~21M parameters. Could this capacity difference contribute materially to the performance gap? If resources allow, please include a capacity-matched Shared-Dreamer baseline (e.g., a ~21M shared world model) or otherwise justify why the comparison remains fair despite the parameter disparity.
> >
> > If the above concerns can be addressed, I will increase my rate.

---

> > > ### Author Response · Authors · 2026-04-07
> > >
> > > **Question 1**
> > >
> > > We appreciate the suggestion to report longer-horizon results. As the reviewer accurately noted, running these evaluations to 1M steps is highly computationally expensive. While completing the full 1M steps before the deadline was physically impossible, we maximized our computational resources to successfully extend the training horizon to **250K steps (exactly 5x our original 50K budget)** for the Proprio Control benchmark. **Furthermore, to provide a holistic evaluation that simultaneously addresses your concern regarding model capacity (which will be detailed in Question 3), we also included the capacity-scaled Shared-Dreamer baselines (21M and 189M) in these extended runs.** The results are summarized in the table below:
> > >
> > > | Metric | PPO | DMPO | TD-MPC2 | DreamerV3 | Shared-Dreamer (12M) | Shared-Dreamer (21M) | Shared-Dreamer (189M) | Ours |
> > > | :--- | :--- | :--- | :--- | :--- | :--- | :--- | :--- | :--- |
> > > | Mean | 229.1 | 684.8 | 809.9 | 757.8 | 344.8 | 352.5 | 358.9 | **822.0** |
> > > | Median | 126.1 | 799.8 | 871.7 | 868.5 | 305.8 | 307.2 | 299.1 | **921.3** |
> > >
> > > As the table demonstrates, under this **5x longer horizon**, our method not only sustains its clear advantage but continues to consistently outperform all strong baselines, including DreamerV3 and TD-MPC2. Importantly, even with 5x more training steps and massively scaled parameters (up to 189M), the shared models still suffer from severe negative transfer and fail to recover asymptotic performance. This substantial extension provides concrete evidence that our observed gains reflect genuinely improved asymptotic performance and successful mitigation of negative transfer, rather than merely faster initial convergence. Furthermore, it explicitly alleviates the concern regarding potentially worse long-run performance.
> > >
> > > **We apologize for the delayed response, as we have been continuously running these intensive experiments to ensure the rigor of our results.** We are currently continuing these extended runs on our servers and will add more longer-horizon results in the revision.
> > >
> > > **Question 2**
> > >
> > > We apologize for omitting this detail. Following the Experience Replay implementation of DreamerV3, we parameterize training via the replay ratio to set a universal update-to-data ratio of 1.0 (identical to the DreamerV3 baseline) across all tasks, thereby avoiding the need for any task-specific hyperparameter tuning. This corresponds to exactly 1 gradient step for each environment interaction step. Consequently, the **50K interaction steps** reported in our experiments entail exactly **50K gradient steps** for both the world model and the actor critic networks, **as they are updated concurrently**. We will explicitly clarify this hyperparameter in the revision.
> > >
> > > **Question 3**
> > >
> > > We appreciate the reviewer raising this important point regarding model capacity. As recently discussed in literature (e.g., Zhai et al., "Stabilizing Visual Reinforcement Learning via Asymmetric Interactive Cooperation", ICCV 2023), simply scaling up model parameters in online reinforcement learning may lead to training instability rather than high performance gains. To directly address your concern, we conducted additional experiments on the Proprio Control benchmark by scaling up the Shared-Dreamer baseline.
> > >
> > > Specifically, we introduced two new baselines: **Shared-Dreamer (21M)** (matching our per-group model) and **Shared-Dreamer (189M)** (matching the total parameter count of our method across all 9 groups). Following the standard scaling protocol of DreamerV3, we constructed these larger baselines by proportionally increasing the base model dimension while keeping the network architecture and all hyperparameters strictly fixed.
> > >
> > > | Metric | DreamerV3 (12M x 18) | Shared-Dreamer (12M x 1) | Shared-Dreamer (21M x 1) | Shared-Dreamer (189M x 1) | Ours (21M x 9) |
> > > | :--- | :--- | :--- | :--- | :--- | :--- |
> > > | Mean | 456.1 | 280.3 | 292.8 | 301.8 | **647.4** |
> > > | Median | 501.5 | 257.0 | 259.6 | 217.9 | **775.7** |
> > >
> > > As shown in the table, simply increasing the capacity of the shared model fails to provide consistent improvements. While the massively scaled Shared-Dreamer (189M) yields a marginal gain in the mean metric, its median performance actually degrades significantly compared to the 12M and 21M baselines. Both Shared-Dreamer (21M) and Shared-Dreamer (189M) still suffer from severe negative transfer, performing much worse than the single-task DreamerV3 baseline.
> > >
> > > This clearly demonstrates that the capacity difference does not materially contribute to the performance gap. The increased parameters in our Stage-2 model (from 12M to 21M) stem specifically from the MoE module, which meaningfully integrates dynamics predictors from other groups. These results confirm that our significant performance gain comes from our carefully designed framework, rather than mere parameter expansion. We will include these capacity-matched baselines and discussions in the revision.

---

### Official Review · Reviewer_BFFG · 2026-03-13

**Soundness:** 3
**Presentation:** 3
**Significance:** 2
**Originality:** 3
**Overall Recommendation:** 4
**Confidence:** 3

**Summary:**

This paper proposes a knowledge-sharing world model for online reinforcement learning. To enable knowledge sharing between tasks while mitigating possible negative transfer and interference, this paper introduces a two-stage learning paradigm. In the first stage, tasks are grouped based on gradient similarity, and world models are trained separately for each group. In the second stage, these models are trained jointly to aggregate knowledge. The method has been evaluated on several benchmarks, including Proprio Control, Visual Control, and Meta-World, and demonstrates effectiveness compared to model-free RL methods and MBRL baselines.

**Compliance With Llm Reviewing Policy:**

Affirmed.

**Final Justification:**

The authors have addressed my main concerns and I will keep my positive rating.

**Key Questions For Authors:**

Please see weaknesses.

**Limitations:**

Yes

**Strengths And Weaknesses:**

**Strengths**
1) The paper is well-motivated and easy to follow.
2) On Proprio Control and Visual Control, the proposed method shows better performance compared with DreamerV3, which validates its effectiveness.
3) The ablation study is rigorously designed, showing that getting rid of negative transfer between tasks is crucial.

**Weaknesses**
1) It would be great if DreamerV3 and the Shared-Dreamer baseline can be also compared on the Meta-World benchmark. For benchmarks such as Proprio Control and Visual Control, different tasks exhibit clear similarities and dissimilarities. However, for robotics tasks like Meta-World, it remains uncertain whether employing a two-stage method yields significant gains.
2) It is widely acknowledged that video world models like Genie 3 can benefit from training from all data sources with one unified model. I’m curious about if the proposed method can be also applied to other formats of world models.
3) Besides the benchmarks presented in this paper, it would be beneficial to study and discuss the performance implications of scaling the total number of tasks.

---

> ### Author Rebuttal · Authors · 2026-03-31
>
> **Weakness 1**
>
> We sincerely thank the reviewer for this constructive suggestion. To demonstrate the effectiveness of our two-stage method on robotics tasks, we have conducted additional experiments on the Meta-World benchmark, with a particular focus on the MT10 suite as an example.
>
> As shown in the table below, our method significantly outperforms both the DreamerV3 and Shared-Dreamer baselines. These results confirm that our approach yields substantial performance gains and remains highly effective in complex robotics domains like Meta-World. We will include these results in the revised manuscript.
>
> | Algorithm | Success Rate |
> | :--- | :--- |
> | DreamerV3 | $0.300 \pm 0.129$ |
> | Shared-Dreamer | $0.264 \pm 0.212$ |
> | **Ours** | $\mathbf{0.517 \pm 0.075}$ |
>
> **Weakness 2**
>
> We appreciate the reviewer highlighting the potential of unified world models like Genie 3. While the lack of an open-source implementation currently prevents us from experimenting directly on Genie 3, we fully agree with the importance of demonstrating generalizability across different formats of world models. To verify this, we integrated our framework with **TD-MPC2**.
>
> It is worth noting the fundamental architectural contrast here: while our primary baseline relies on an RSSM architecture that learns via observation reconstruction, TD-MPC2 represents a completely different format. It employs an **implicit (decoder-free) world model** and performs local trajectory optimization directly in the latent space.
>
> We evaluated this new variant on the Proprio Control benchmark. To ensure a comprehensive comparison, we included two baselines: the standard online **TD-MPC2** (where a separate model is trained independently for each task) and a **Shared-TD-MPC2** (which simply forces all tasks to share a single, unified online world model).
>
> As illustrated in the table below, the Shared-TD-MPC2 approach suffers a severe performance drop compared to the single-task baseline. This highlights that naively mixing multi-task data is ineffective and triggers severe negative transfer, even in implicit decoder-free architectures. Conversely, applying our framework to TD-MPC2 safely and effectively leverages cross-task data, leading to clear performance improvements over the single-task baseline.
>
> | Metric | TD-MPC2 | Shared-TD-MPC2 | Ours (TD-MPC2 version) |
> | :--- | :--- | :--- | :--- |
> | Mean | 619.1 | 220.7 | 639.4 |
> | Median | 750.6 | 152.7 | 783.9 |
>
> **Importantly, we integrated our framework with TD-MPC2 in a direct manner without conducting any architecture-specific hyperparameter tuning.** The fact that our method achieves these clear improvements under such a zero-tuning setting demonstrates that TaskLoom is not strictly limited to RSSM formats, but can be successfully and robustly generalized to other types of world models. We will add a discussion on this generalizability in the revised manuscript.
>
>
> **Weakness 3**
>
> To investigate the performance implications of scaling the total number of tasks, we conducted an ablation study within the Proprio Control setting. We trained our model using varying total task counts (6, 12, and 18 tasks). Specifically, these correspond to the first 6, first 12, and all 18 tasks from the following ordered list: Acrobot Swingup, Cup Catch, Pendulum Swingup, Finger Spin, Finger Turn Easy, Finger Turn Hard (which serve as our fixed subset of 6 base tasks), Reacher Easy, Reacher Hard, Cartpole Balance Sparse, Cartpole Swingup Sparse, Cartpole Balance, Cartpole Swingup, Walker Run, Walker Walk, Hopper Hop, Hopper Stand, Cheetah Run, and Walker Stand. We tracked the performance of the fixed 6 base tasks across all three settings.
>
> As shown in the table below, the performance on the base tasks consistently improves as the total number of training tasks increases. This evidence clearly indicates that our method can effectively scale up and positively leverage a broader diversity of training tasks to enhance overall performance.
>
> | Metric | 6 tasks | 12 tasks | 18 tasks |
> | :--- | :--- | :--- | :--- |
> | Mean | 586.7 | 591.1 | **613.3** |
> | Median | 584.5 | 591.0 | **605.6** |

---

> > ### Author Rebuttal · Reviewer_BFFG · 2026-04-01
> >
> > The authors have addressed my main concerns and I will keep my positive rating.

---

> > > ### Author Response · Authors · 2026-04-02
> > >
> > > We sincerely thank the reviewer for evaluating our rebuttal and for the continued positive support. We are very glad that our responses have fully addressed your concerns, and we deeply appreciate your constructive feedback.

---

### Decision · Program_Chairs · 2026-04-30

**Decision:**

Accept (regular)

**Comment:**

While the reviewers initially had concerns regarding generalization across benchmarks, architectural dependence (especially beyond RSSM-based world models), and the clarity of certain training and evaluation details, these issues were largely addressed during the rebuttal, and all reviewers now recommend Weak Accept. Overall, the paper shows solid empirical results with consistent improvements across benchmarks and strong ablation support.

Please address the issues raised in the Final Justifications, and ensure that the final version includes the clarifications and additional details promised during the rebuttal.